# A quantification method for heat-decomposable methylglyoxal oligomers and its application on 1,3,5-trimethylbenzene SOA

Maria Rodigast, Anke Mutzel, and Hartmut Herrmann*

*Leibniz Institute for Tropospheric Research (TROPOS), Atmospheric Chemistry Dept. (ACD), Permoserstr. 15, D-04318 Leipzig, Germany*

*Corresponding author. Tel: +49-341-2717-7024; fax: +49-341-271799-7024.

*E-mail address:* herrmann@tropos.de

For submission to: Atmospheric Chemistry and Physics

First submitted on July 02$^{nd}$ 2016, revised August 09$^{th}$, 2016

**Abstract**
Methylglyoxal forms oligomeric compounds in the atmospheric aqueous particle phase, which could establish
a significant contribution to the formation of aqueous secondary organic aerosol (aqSOA). Thus-far, no
suitable method for the quantification of methylglyoxal oligomers is available despite the great effort spent for
structure elucidation. In the present study a simplified method was developed to quantify heat-decomposable
methylglyoxal oligomers as a sum parameter. The method is based on the thermal decomposition of oligomers
into methylglyoxal monomers. Formed methylglyoxal monomers were detected using PFBHA (o-(2,3,4,5,6-
pentafluorobenzyl)hydroxylamine hydrochloride) derivatisation and gas chromatography/mass spectrometry
(GC/MS) analysis. The method development was focused on the heating time (varied between 15 and 48
hours), pH during the heating process (pH = 1 - 7), and heating temperature (50°C, 100°C). The optimised
values of these method parameters are presented.
The developed method was applied to quantify heat-decomposable methylglyoxal oligomers formed during
the OH-radical oxidation of 1,3,5-trimethylbenzene (TMB) in the Leipziger aerosol chamber (LEAK).
Oligomer formation was investigated as a function of seed particle acidity and relative humidity. A fraction of
heat-decomposable methylglyoxal oligomers of up to 8% in the produced organic particle mass was found,
highlighting the importance of those oligomers formed solely by methylglyoxal for SOA formation. Overall,
the present study provides a new and suitable method for quantification of heat-decomposable methylglyoxal
oligomers in the aqueous particle phase.

## 1. Introduction

Aromatic compounds represent a large fraction of the emitted hydrocarbons, contributing up to 52% to the total non-methane hydrocarbon mass at an industrial dominated site in China (Liu et al., 2008). One of these aromatic compounds is 1,3,5-trimethylbenzene (TMB), which was measured in the gas phase in concentrations ranging from 0.7 to 40.6 µg m$^{-3}$ (Gee and Sollars, 1998; Khoder, 2007). The gas-phase oxidation of 1,3,5-TMB leads to low-volatility oxidation products, which partition into the particle phase and form secondary organic aerosol (SOA). Oxidation products of 1,3,5-TMB were investigated in a number of literature studies (e. g. Huang et al., 2015; Baltensperger et al., 2005; Kalberer et al., 2004; Kalberer et al., 2006; Paulsen et al., 2005; Healy et al., 2008; Cocker et al., 2001; Smith et al., 1999; Metzger et al., 2008; Wyche et al., 2009; Yu et al., 1997). Methylglyoxal was found as one of the most important oxidation product (Metzger et al., 2008; Healy et al., 2008; Cocker et al., 2001; Smith et al., 1999; Wyche et al., 2009; Baltensperger et al., 2005; Rickard et al., 2010; Kalberer et al., 2004; Yu et al., 1997; Kleindienst et al., 1999; Müller et al., 2012; Nishino et al., 2010; Hamilton et al., 2003; Tuazon et al., 1986; Bandow and Washida, 1985; Lim and Turpin, 2015) contributing with a fraction of up to 2% to the particle mass (Healy et al., 2008; Cocker et al., 2001). Methylglyoxal has often been described to form oligomeric compounds in the aqueous particle phase (see, e.g. Herrmann et al., 2015 for an overview; De Haan et al., 2009; Kalberer et al., 2004; Loeffler et al., 2006; Zhao et al., 2006; Sareen et al., 2010; Altieri et al., 2008), which are supposed to play an important role in the formation of aqueous secondary organic aerosols (aqSOA; e.g. Kalberer et al., 2004).

In general, oligomeric compounds can be formed in the aqueous particle phase through aldol condensation (e. g. Tilgner and Herrmann, 2010; Sareen et al., 2010; Sedehi et al., 2013; Krizner et al., 2009; Barsanti and Pankow, 2005; De Haan et al., 2009; Yasmeen et al., 2010), acetal/hemiacetal formation (Kalberer et al., 2004 and Yasmeen et al., 2010), esterification (Altieri et al., 2008; Sato et al., 2012; Tan et al., 2010; De Haan et al., 2011; Sedehi et al., 2013), imine formation (Altieri et al., 2008; Sato et al., 2012; Tan et al., 2010; De Haan et al., 2011; Sedehi et al., 2013), hydrolysis of epoxides (Paulot et al., 2009; Surratt et al., 2010), polymerisation, and radical – radical reactions (Schaefer et al., 2015; Tan et al., 2012; Lim et al., 2013).

During the last decade, huge efforts were undertaken to detect and identify oligomeric compounds. As it can be seen in Table 1, a number of mass spectrometric methods were used including (matrix assisted) laser desorption/ionisation mass spectrometry (MALDI-MS, LDI-MS), Fourier transform ion cyclotron resonance mass spectrometry (FT-ICR-MS), electrospray ionisation mass spectrometry and electrospray ionisation tandem mass spectrometry (ESI/MS, ESI/MS/MS). In addition, spectroscopic methods like UV/Vis (ultraviolet-visible spectroscopy), FTIR (Fourier transform infrared spectroscopy), and NMR (nuclear magnetic resonance spectroscopy) analysis were used for identification.

Despite the past effort for structure elucidation of oligomeric compounds a suitable quantification method is not available. Mostly, an overall contribution of oligomers to the particle mass was determined using, e.g., a volatility tandem differential mobility analyser (VTDMA). Kalberer et al. (2004) determined an oligomer contribution of 50% to the particle mass formed by the photooxidation of 1,3,5-TMB. In a further experiment, oligomer mass fractions of 80% and 90% were determined with a VTDMA-based approach for 1,3,5-TMB and α-pinene (Kalberer et al., 2006). Alfarra et al. (2006) investigated the photooxidation of 1,3,5-TMB and found an increase of the oligomer fraction of 3.1 and 3.7% hour$^{-1}$. A particulate oligomer fraction of 50% was reported for 1,3,5-TMB and α-pinene by Baltensperger et al. (2005). Dommen et al. (2006) detected

a contribution of oligomers to the organic particle mass increasing from 27% to 44% in the first 5 hours of the
photooxidation of isoprene. Nguyen et al. (2011) investigated oligomers from isoprene photooxidation with
ESI-MS and nano-DESI-MS (nanospray desorption electrospray ionisation) connected to a high resolution
linear ion trap (LTQ-) orbitrap. They calculated an oligomer fraction of 80 – 90%.

De Haan et al. (2009) estimated the oligomer fraction formed by methylglyoxal in the aqueous phase

with NMR concluding 37% of methylglyoxal are dimers and oligomers. In another approach it was estimated
that after 4 days in aqueous particles containing amino acids, 15% of the carbonyl compounds are oligomers
(Noziere et al., 2007). Contrary, with ammonium sulfate particles 30% of the carbonyl compounds are
converted into oligomers (Noziere et al., 2007). These estimations are based on the rate constants for oligomer
formation which were determined in the study by Noziere et al. (2007). Besides these estimations,
quantification of oligomeric compounds was also conducted using surrogate compounds (Surratt et al., 2006,
Zappoli et al., 1999, and Gao et al., 2004) or synthesised authentic standards (Birdsall et al., 2013).

In summary, a variety of methods exists which quantify the fraction of oligomeric compounds derived

from methylglyoxal, but the results are contradicting due to the lack of a suitable method for quantification
and second, due to different reaction conditions used in the studies. Thus, the present study presents a
fundamental approach for a reliable quantification of methylglyoxal oligomers in laboratory-generated SOA.
The method is applicable for any oligomeric compounds, which can be decomposed into methylglyoxal
monomers during the heating process at a temperature of 100 °C. As the oligomerisation mechanisms leading
to the quantified oligomeric compounds are not known, it cannot be specified if the oligomers are reversibly
or irreversibly formed. In addition, it cannot be excluded that there exist oligomers, which are not
decomposable into their methylglyoxal monomers through the heating process. Thus, the quantified oligomers
are termed as heat-decomposable methylglyoxal oligomers.

**2. Experimental**
2.1 Chemicals and standards
1,3,5-TMB ($\geq$ 99.8%), hydrochloric acid (37%), tetramethylethylene (99%), and sodium hydroxide
(50 – 52%) were obtained from Sigma-Aldrich (Hamburg, Germany). $O$-(2,3,4,5,6-pentafluorobenzyl)-
hydroxylamine hydrochloride ($\geq$99%), methylglyoxal (40% in water), and ammonium hydrogensulfate (98%)
were purchased from Fluka (Hamburg, Germany). Sulfuric acid (98%) was obtained from Merck KGaA
(Darmstadt, Germany). Dichloromethane (Chromasolv 99.8%) was obtained from Riedel-de Haen (Seelze,
Germany) and ammonium sulfate (99.5%) was purchased from Carl Roth (Karlsruhe, Germany). Ultrapure
water was used to prepare the seed particle solutions, the authentic standards, and to extract the filter samples
(Milli-Q gradient A 10, 18.2 M$\Omega$ cm, 3 ppb TOC, Millipore, USA).

2.2 Chamber experiments
The OH-radical oxidation of 1,3,5-TMB was investigated in the LEipziger AerosolKammer (LEAK). A
detailed description of the aerosol chamber can be found elsewhere (Mutzel et al., 2016). The conditions of
the experimental runs are summarised in Table 2. The experiments were conducted in the presence of
ammonium bisulfate particles or ammonium sulfate particles mixed with sulfuric acid to achieve different seed
acidities. In order to investigate OH-radical oxidation of 1,3,5-TMB at low NO$_x$ levels (< 1 ppb) and under
dark conditions the ozonolysis of tetramethylethylene (TME) was used as OH-radical source (Berndt and
Böge, 2006). The cycloaddition of ozone to TME yields a primary ozonide, which reacts further and forms a
stabilised Criegee Intermediate (sCI). The sCI decomposes via the hydroperoxide channel, leading to the
formation of OH radicals (Gutbrod et al., 1996) with a yield of $0.92 \pm 0.08$ (Berndt and Böge, 2006). $O_3$ was
produced by UV irradiation of $O_2$ with an $O_2$ flow rate of 5 L minute$^{-1}$. It was injected at the beginning of the
experiments and $\approx 26$ ppbv of TME was introduced into the aerosol chamber in steps of 15 minutes. 1,3,5-
TMB ($\approx 92$ ppb) was injected into the aerosol chamber using a microliter-syringe. The oxidation of 1,3,5-TMB
was studied at relative humidities (RH) between $\approx 0\%$ and 75% adjusted by flushing the aerosol chamber with
humid or dry air. The consumption of the precursor compound ($\Delta$HC) was monitored over a reaction time of
90 minutes with a proton-transfer-reaction time-of-flight mass spectrometer (PTR-TOF MS; 8000, IONICON
Analytik, Innsbruck, Germany). The volume size distribution of the seed particles was measured with a
scanning mobility particle sizer (SMPS; 3010; TSI, USA). An average density of 1 g cm$^{-3}$ was used to calculate
the increase of the organic particle mass ($\Delta$M). To collect the particle phase after the experiments, 1.2 m³ of
the chamber volume was sampled on a PTFE filter (borosilicate glass fiber filter coated with fluorocarbon, 47
mm in diameter, PALLFLEX T60A20, PALL, NY, USA) connected to a XAD-4 coated denuder (URG-2000-
30B5, URG Corporation, Chapel Hill, NC, USA; Kahnt et al., 2011) to avoid artefacts caused by adsorption
of gas-phase organic compounds onto the filter.

2.3 Sample preparation
For method development, PTFE filter samples from aerosol chamber experiments were used. The following
method parameters were investigated: heating time, pH during the heating process, and heating temperature
(Table 3). Filter samples from the same experiments were used for the optimisation of the respective method
parameters.

2.3.1 Filter extraction
Two halves of the PTFE filters were cut into small pieces. Each filter half was extracted separately with 1 mL
$H_2O$ for 30 minutes using an orbital shaker (700 rpm, revolutions per minutes). They were shaken again
separately with 1 mL $H_2O$ for 30 minutes and flushed at the end of the procedure with 1 mL $H_2O$ resulting in
two 3 mL extracts. Notably, the extraction efficiency was not investigated in the present study, thus it is not
known. Water was used as extracting reagent, because organic solvents like methanol and acetonitrile have
lower boiling points than water, thus lower heating temperatures can be applied for the decomposition of the
oligomers. Besides this, organic solvents like methanol and acetonitrile are miscible with dichloromethane,
thus an extraction of the derivatised methylglyoxal with dichloromethane prior the injection into GC/MS would
not be possible. The extract of one half of the filter was used for oligomer measurements (extract 1) and with
the second one methylglyoxal monomers were quantified which were not a building block of oligomers
(extract 2). A detailed description of the derivatisation procedure can be found in Rodigast et al. (2015).

2.3.2 Derivatisation procedure
*Extract 1 – heat-decomposable methylglyoxal oligomers*
For quantification of heat-decomposable methylglyoxal oligomers the extract was acidified and heated to
decompose the oligomeric bonds. The pH was adjusted with hydrochloric acid (37%) or sodium hydroxide
(1 mol L$^{-1}$) to pH = 1, 3, 5, and 7 while heating temperatures of 50°C and 100°C were investigated. For the
derivatisation of the formed monomeric methylglyoxal, 300 µL of *o*-(2,3,4,5,6-
pentafluorobenzyl)hydroxylamine hydrochloride (PFBHA, 5 mg mL$^{-1}$) was added to the sample solution after
2 minutes of the heating process. Different heating times were tested varying between 15 hours and 48 hours.
After the derivatisation was complete, the extracts were allowed to cool down to room temperature.

*Extract 2 – Methylglyoxal monomer*
The second half of the filters was used to quantify monomeric methylglyoxal including also singly and doubly
hydrated methylglyoxal, which can be hydrolysed at room temperature into monomers. The filters were
prepared according to the method described by Rodigast et al. (2015), thus 300 µL of an aqueous PFBHA
solution (5 mg mL$^{-1}$) was added to the filter extracts. After a derivatisation time of 24 hours at room
temperature the derivatised methylglyoxal monomers were extracted.

2.3.3 Extraction for GC/MS analysis
After derivatisation of both filter extracts (extract 1 and 2), derivatised methylglyoxal was extracted at pH = 1
for 30 minutes with 250 µL of dichloromethane using an orbital shaker (1500 rpm; Rodigast et al., 2015). 1 µL
of the organic phase was injected into GC/MS for analysis. The measurements were repeated for three times
to ensure reliable GC/MS signals.
For quantification, a 5-point calibration was performed at the beginning of each chromatographic run using a
standard solution of methylglyoxal in a concentration range of 0.13 to 8 µmol L$^{-1}$.

2.4 Instrumentation
The samples were analysed using a GC System (6890 Series Agilent Technologies, Frankfurt, Germany)
coupled with an electron ionisation quadrupole mass spectrometer in splitless mode with an inlet temperature
of 250°C (Agilent 5973 Network mass selective detector, Frankfurt, Germany). The derivatives were separated
with a HP-5MS UI column (Agilent J & W GC columns, 30 m × 0.25 mm × 0.25 µm) using the following
temperature program: 50°C isothermal for 2 minutes and elevated to 230°C (10 °C minute$^{-1}$). The temperature
of 230°C was held constant for 1 minute and ended with 320°C for 10 minutes, thus the method has a run time
of 36 minutes.

**3. Results**
Hastings et al. (2005) investigated the influence of the temperature of the GC inlet on the detection of
oligomeric compounds. These authors concluded that oligomers decompose into monomer building blocks at
higher inlet temperatures (≥ 120°C) which caused problems for oligomer quantification. In the present study a
quantification method is proposed to decompose heat-decomposable oligomers into methylglyoxal monomers
due to heating, acidification and PFBHA derivatisation prior GC/MS injection.
(E) and (Z) isomers of methylglyoxal were formed during PFBHA derivatisation resulting in two peaks in the
GC/MS chromatogram. For quantification, the sum of these peaks was used to avoid an over- or
underestimation of methylglyoxal due to variations of the isomer peak ratio during the heating process.

**3.1 Method development**
*Influence of heating time*
The influence of the heating time was examined with PTFE filters which were sampled after OH-radical
oxidation of 1,3,5-TMB at RH = 75% in the presence of $NH_4HSO_4$ seed particles (experiment #3). To
investigate the effect of the heating time on the decomposition of the heat-decomposable oligomeric
compounds, the aqueous filter extracts (extract 1) were acidified to pH = 1 and heated to 100°C for
15 – 48 hours. The results were compared to the unheated aqueous filter extracts (extract 2) to determine the
increase of methylglyoxal concentration due to decomposition of the heat-decomposable oligomers.
Additionally, a 6.25 µmol $L^{-1}$ standard solution of methylglyoxal was acidified and heated for different times
to exclude an effect of the heating process on the derivatisation. The results are illustrated in Fig. 1a.
The highest methylglyoxal concentration can be found after a heating time of 24 hours. The methylglyoxal
concentration was about six times higher (c = 1.82 ± 0.14 µmol $L^{-1}$) in comparison to the unheated filter extract
(c = 0.29 ± 0.01 µmol $L^{-1}$). To exclude that the higher methylglyoxal concentrations were only a result of a
better PFBHA derivatisation during heating, a methylglyoxal standard solution was also heated for 24 hours
(Fig. 1b). A methylglyoxal concentration of c = 5.32 ± 0.05 µmol $L^{-1}$ was found, which corresponds to a
recovery of ≈ 85%. Thus, an effect of the heating process on the derivatisation can be excluded indicating that
the higher methylglyoxal concentration was caused by decomposition of heat-decomposable oligomers into
monomers.
One filter extract was heated for 15 hours and allowed to stand at room temperature for 9 hours
(sample a) to reach a total derivatisation time of 24 hours (as it was optimised for PFBHA derivatisation by
Rodigast et al., 2015). To exclude reoligomerisation processes of methylglyoxal in sample a, one filter extract
was heated for 15 hours and measured directly after the heating process (sample b). As it can be seen in Fig. 1a
both filter samples (sample a and b) showed lower methylglyoxal concentrations than after heating for
24 hours. The lower methylglyoxal concentration of sample b might be caused by an incomplete derivatisation
due to the immediate measurement of the filter extract after 15 hours heating time. In comparison, the lower
concentration in sample a might be caused by reoligomerisation of methylglyoxal.
To probe this hypothesis a 6.25 µmol $L^{-1}$ methylglyoxal standard was heated for 15 hours and
measured immediately (like sample a) or, alternatively, was allowed to stand at room temperature for 9 hours
(like sample b). In Fig. 1b a lower methylglyoxal concentration can be observed for the immediately measured
sample  (3.11 ± 0.20 µmol $L^{-1}$)  compared  to  the  sample  after  9  hours  at  room  temperature
(5.84 ± 0.27 µmol $L^{-1}$). Thus, it can be concluded that a derivatisation time of 24 hours is needed for a complete
derivatisation despite the heating process. This supports the hypothesis that methylglyoxal monomers were not
completely derivatised, if the filter sample was heated for 15 hours and directly measured (sample b). Based
on the incomplete derivatisation after 15 hours heating time, methylglyoxal monomers are able to react again
under oligomer formation during the 9 hours at room temperature. The rate constants are reported to be
k = $5 \times 10^{-6}$ $M^{-1}$ minutes$^{-1}$ for ammonium ion catalysed and k ≤ $1 \times 10^{-3}$ $M^{-1}$ minutes$^{-1}$ for $H_3O^+$ catalysed aldol
reaction (Sareen et al., 2010). Naturally, both of these ions are present in the aqueous filter extract. Despite the
oligomerisation of methylglyoxal monomers during the 9 hours at room temperature, the derivatisation
proceeds as well during this time leading to higher methylglyoxal concentrations in sample b than in sample
a, which was directly measured after 15 hours heating (Fig. 1a).

Longer heating times than 24 hours (30 and 48 hours) led to lower methylglyoxal concentrations in
the filter samples as well. A possible explanation might be the decomposition of the derivatised compound
during the long heating process. As no decrease of the concentration was observed in the methylglyoxal
standard solution (Fig. 1b) the loss of the derivatisation group is unlikely as a reason for the lower
concentrations. Thus, it can be speculated that the low methylglyoxal concentrations in the filter samples are
a result of further reactions with particle-phase species, which do not exist in the standard samples. Based on
the outlined results, a heating time of 24 hours was chosen.

The pH during the heating process was investigated as well.


*Influence of pH*
The effect of the pH was examined with PTFE filters, which were sampled after OH-radical oxidation of 1,3,5-
TMB at RH = 50% in the presence of $NH_4HSO_4$ particles (experiment #2). The pH was varied between pH = 1
and pH = 7.
As it can be seen in Fig. 2a the highest methylglyoxal concentration can be found at pH = 1. The methylglyoxal
concentration was about two times higher at pH = 1 (c = 1.01 ± 0.11 µmol $L^{-1}$) compared to the filter extract,
which was neither heated nor acidified (c = 0.45 ± 0.01 µmol $L^{-1}$). An increasing pH leads to a lower
methylglyoxal concentration, which can be observed for filter samples (Fig. 2a) as well as for the
methylglyoxal standard solution (Fig. 2b). As this was observed for both types of samples it appears that the
pH influences the derivatisation and/or the oligomer decomposition. No influence of the pH on the PFBHA
derivatisation reaction was reported by Rodigast et al. (2015) indicating the effect of the pH is connected to
thermal decomposition of the heat-decomposable oligomeric compounds. In summary, based on these results
pH = 1 was used.

*Influence of heating temperature*
The effect of the heating temperature was examined with filter samples of experiment #2. The heating
temperature was varied between 50°C and 100°C and the filter extracts were heated for 24 hours at pH = 1. A
temperature above 100°C cannot be used to avoid evaporation of water and/or target compounds. Fig. 3a shows
the influence of the temperature on the decomposition of the heat-decomposable methylglyoxal oligomers into
monomers. Higher concentration of methylglyoxal can be detected with higher temperature. The results
illustrated in Fig. 3a indicate that a higher temperature than 50°C is needed to decompose the oligomeric
compounds. In comparison to the filter, which was neither acidified nor heated, the concentration increased by
a factor of two if the extract was heated to 100°C. Fig. 3b shows no significant influence of the temperature
on the methylglyoxal standard solution. Thus, an influence of the heating temperature on the derivatisation
procedure can be excluded.

Based on these results, the PTFE filter extracts from the aerosol chamber experiments were acidified
to pH = 1 and heated for 24 hours to 100°C to decompose heat-decomposable oligomeric compounds into
methylglyoxal. According to the literature studies other carbonyl compounds can be expected as particle-phase
products, e.g., propionaldehyde (Cocker et al., 2001), glyoxal (Cocker et al., 2001; Huang et al., 2015), 2-
methyl-4-oxo-2-pentenal (Healy et al., 2008; Huang et al., 2014), glycolaldehyde (Cocker et al., 2001) and
3,5-dimethylbenzaldehyde (Huang et al., 2014). Noticeably, no carbonyl compounds other than methylglyoxal
were identified, which showed an increase after thermal decomposition.

The developed quantification method was afterwards applied to laboratory-generated SOA formed

during further oxidation experiments of 1,3,5-TMB to investigate the influence of seed particle acidity and
relative humidity on the oligomer content.

### 3.2 SOA yield and growth curves of 1,3,5-TMB oxidation

SOA formation of 1,3,5-TMB was investigated in a number of literature studies mostly in the presence of $NO_x$
and under variation of the hydrocarbon to $NO_x$ ratio ([HC]/[$NO_x$] ratio). Healy et al. (2008) determined SOA
yields ($Y_{SOA}$) of 1,3,5-TMB photooxidation ranging from 4.5 to 8.3%. Further studies determined $Y_{SOA}$
between 0.29% and 15.6% (Table 4) under variation of the [HC]/[$NO_x$] ratio concluding SOA formation is
enhanced at low $NO_x$ mixing ratios (Sato et al., 2012; Kleindienst et al., 1999; Baltensperger et al., 2005;
Wyche et al., 2009; Odum et al., 1997; Paulsen et al., 2005; Cocker et al., 2001). Only Cao and Jang (2007)
investigated SOA yields in the absence of $NO_x$ and reported values between 7.1 and 13.8%. The SOA yields
were also determined in the present study for all conducted experiments based on the ratio of $\Delta M$ to $\Delta HC$
(Table 2). Notably, the SOA yields of 1,3,5-TMB are not corrected for the wall loss to the surface of the aerosol
chamber, which might possibly lead to an underestimation of the reported SOA yields. $Y_{SOA}$ varied between 4
and 7% dependent on reaction conditions and is in good agreement to literature values.

For a further investigation of SOA-formation processes of 1,3,5-TMB, Fig. 4a illustrates the

dependency between the consumption of 1,3,5-TMB ($\Delta HC$) and the produced organic particle mass ($\Delta M$).
Particle growth started directly after the experiment was initialised indicating that the oxidation leads
immediately to the formation of condensable products as first-generation oxidation products. These products
condense on the pre-existing seed particles resulting in the immediate particle growth observed in Fig. 4a.
Differences of the growth curves in dependence on the seed particles ($NH_4HSO_4$ and $(NH_4)_2SO_4/H_2SO_4$) were
not observed concluding that the seed particle acidity (Table 5) has no influence on the SOA formation of
1,3,5-TMB. Cao and Jang (2007) found also only a small influence of seed particle acidity on SOA formation.

Fig. 4a showed great differences in the growth curves under variation of RH. The RH value can have

an influence on the phase state of the particles which, in turn, has an effect on the partitioning of the compounds
into the particles as well as on  particle-phase reactions (Ziemann, 2010; Saukko et al., 2012). The
deliquescence RH of $(NH_4)_2SO_4$ and $NH_4HSO_4$ seed particles are known from literature to be 79% and 39%,
respectively (Cziczo et al., 1997). Thus, pure $NH_4HSO_4$ seed particles are liquid at the applied RH values of
50% and 75% and solid at RH = 0% while $(NH_4)_2SO_4$ seed particles are solid over the whole RH range.
Nevertheless, in the present study the particles are a mixture of inorganic and partitioned organic compounds.
Organic compounds might change the phase state of the particles (Bertram et al., 2011) due to an influence on
the deliquescence point (Andrews and Larson, 1993; Lightstone et al., 2000) as well as the hygroscopic
behaviour of the particles (Lightstone et al., 2000; Prenni et al., 2003; Chen and Lee, 1999). Virtanen et al.
(2010) postulated that particles are in an amorphous solid state if oligomeric compounds are present in the
particles. Thus, it can be speculated that the particles in the present study containing a fraction of up to 8% of
oligomeric compounds might be in an amorphous solid phase state. It was assumed, that the further reactions
in the particle phase might be inhibited in solid particles (Saukko et al., 2012) thus, further oligomerisation or
other reactions might become less effective after a certain fraction of oligomers exist in the particles. In
addition, the partitioning of methylglyoxal monomers can be inhibited into solid particles (Saukko et al., 2012),
which then might also lead to lower oligomer fractions in SOA. Thus, it might be possible that the phase state
influences $\Delta M$ and the SOA yields. $\Delta M$ is the highest at RH = 0% ($\Delta M$ = 18.1 – 19.7 µg m$^{-3}$) whereas $\Delta M$ is
the lowest under humid conditions (RH = 50% and 75%, $\Delta M$ = 11.3 – 11.7 µg m$^{-3}$ and 13.9 – 14.2 µg m$^{-3}$).
Due to the variation of RH in the aerosol chamber the liquid water content (LWC) of the particles is changing
(Table 5). The LWC was calculated using model II from the extended aerosol thermodynamic model (E-AIM;
Clegg et al., 1998). With increasing RH the LWC of the seed particle increases as well. The LWC of the seed
particles influences i) the partitioning of the compounds from the gas phase into the particle phase and ii) the
formation and/or further reaction in the particle phase (Zuend et al., 2010; Cocker et al., 2001; Seinfeld et al.,
2001; Fick et al., 2003). These two effects might influence the SOA formation under different relative
humidities.

An effect can also be seen in Fig. 4b. The SOA formation is enhanced at RH = 0% leading to the

highest SOA yields of $Y_{SOA} \approx 7$% for both seed particles. Higher RH values resulted in lower $Y_{SOA}$ between 4
and 5%. These findings are in good agreement with the study by Cao and Jang (2007), which observed lower
$Y_{SOA}$ values at elevated RH.

The influence of RH on SOA formation is controversial (Hennigan et al., 2008; Fick et al., 2003;

Edney et al., 2000; Saxena and Hildemann, 1996; Baker et al., 2001; Hasson et al., 2001, Cocker et al., 2001).
Edney et al. (2000) and Seinfeld et al. (2001) reported an enhanced SOA formation of hydrophilic compounds
under humid conditions and a lowered SOA formation of hydrophobic compounds. This is also supported by
Saxena and Hildemann (1996), which found an enhanced partitioning of organic compounds with several
hydroxyl groups at higher LWCs of the particles. This might lead to the conclusion that the OH-radical
oxidation of 1,3,5-TMB results in the formation of hydrophobic compounds which showed an enhanced
partitioning under dry conditions. Additionally, it can be speculated that the formation of oligomeric
compounds can be enhanced at lower RH values resulting in higher $Y_{SOA}$ due to the increasing conversion of
the monomeric building blocks and their enhanced partitioning into the particle phase.

**3.3 Particulate methylglyoxal**
Methylglyoxal is reported in the literature as an important oxidation product of 1,3,5-TMB (Metzger et al.,
2008; Healy et al., 2008; Cocker et al., 2001; Smith et al., 1999; Wyche et al., 2009; Baltensperger et al., 2005;
Rickard et al., 2010; Kalberer et al., 2004; Yu et al., 1997; Kleindienst et al., 1999; Müller et al., 2012; Nishino
et al., 2010; Hamilton et al., 2003; Tuazon et al., 1986; Bandow and Washida, 1985) with yields in the particle
phase between 0.7 and 2%.

The fraction of methylglyoxal in the particle phase in dependency on the reaction conditions is shown

in Fig. 5a, with resulting fractions between $\approx 0.6$% and $\approx 2.2$%. With increasing RH the fraction decreases for
both seed particles. Methylglyoxal has the highest fraction under dry conditions (1.73 $\pm$ 0.20% and
2.17 $\pm$ 0.20% for $NH_4HSO_4$ and $(NH_4)_2SO_4/H_2SO_4$) and with $(NH_4)_2SO_4/H_2SO_4$ seed particles.

Healy et al. (2008) measured a contribution of methylglyoxal to SOA mass of $2.06 \pm 0.08\%$ from the

photooxidation of 1,3,5-TMB in the presence of $NO_x$ at RH = 50%. In the present study a methylglyoxal
fraction of $1.24 \pm 0.04\%$ for $NH_4HSO_4$ seed particles and $0.80 \pm 0.08\%$ for $(NH_4)_2SO_4/H_2SO_4$ seed particles
was determined at RH = 50%. Thus, the contribution is slightly lower than measured by Healy et al. (2008).
In comparison, Cocker et al., 2001 measured a particulate fraction of 0.72%.

The dependency of particulate methylglyoxal on RH could be a result of the influence of RH on the

partitioning from the gas- into the particle phase or on further reactions in the particle phase forming oligomers.
The formation of oligomeric compounds from methylglyoxal has been investigated in a number of studies
(e. g. De Haan et al., 2009; Kalberer et al., 2004; Loeffler et al., 2006; Zhao et al., 2006; Sareen et al., 2010;
Altieri et al., 2008).

**3.4 Heat-decomposable methylglyoxal oligomers**
A method was developed to determine the contribution of heat-decomposable methylglyoxal oligomers to the
produced organic particle mass ΔM. The method is based on the thermal decomposition of the heat-
decomposable methylglyoxal oligomers into monomers. Thus, the concentration of monomeric methylglyoxal
was determined prior and after thermal decomposition. The concentrations were converted into the fraction of
methylglyoxal oligomers of ΔM using the molar mass of methylglyoxal (Mw = 72.06 g mol$^{-1}$). An oligomer
fraction of $\approx 2$ up to $\approx 8\%$ was determined.

Fig. 5b shows the dependency of the detected heat-decomposable methylglyoxal oligomers on the

relative humidity with $NH_4HSO_4$ and $(NH_4)_2SO_4/H_2SO_4$ seed particles. In the presence of $NH_4HSO_4$ seed
particles the highest oligomer fraction $(8.2 \pm 0.7\%)$ can be observed with RH = 0% whereas in the presence of
$(NH_4)_2SO_4/H_2SO_4$ seed particles the oligomer fraction is the lowest $(2.1 \pm 0.4\%)$ under dry conditions
(RH = 0%). A possible explanation for the opposite trend of the oligomer fractions with RH between $NH_4HSO_4$
and $(NH_4)_2SO_4/H_2SO_4$ seed particles could be different oligomer formation mechanism caused by different
seed particle acidity. The type of accretion reaction might change with pH (Yasmeen et al., 2010). In Table 5
the pH of the seed particles was calculated with E-AIM. $NH_4HSO_4$ particles have pH = 0.1 and 1.2 at
RH = 50% and 75%. In comparison $(NH_4)_2SO_4/H_2SO_4$ seed particle are less acidic (pH = 4.0 at RH = 50% and
pH = 4.2 at RH = 75%).

It was postulated by Yasmeen et al. (2010) that a lower pH (pH < 3.5) favor acetal/hemiacetal

formation whereas at high pH (pH = 4 - 5) aldol condensation are more relevant. This has been supported by
Sedehi et al. (2013) and Sareen et al. (2010).

Thus, in the presence of strong acidic $NH_4HSO_4$ seed particles acetal/hemiacetal formation might be

the favored oligomer formation mechanism. Oligomerisation via acetal/hemiacetal formation occurs under a
reversible water loss (Yasmeen et al., 2010). As higher RH values in the aerosol chamber LEAK leads to higher
LWCs of the seed particles (Table 5) the chemical equilibrium of the reaction shifts towards the precursor
compound resulting in a lower methylglyoxal oligomer fraction (Kalberer et al., 2004; Liggio et al., 2005). In
addition, the pH of $NH_4HSO_4$ particles decreases with decreasing RH (Table 5) thus acid-catalysed
acetal/hemiacetal formation might be enhanced under dry conditions due to a lower pH (Liggio et al., 2005).

In the presence of $(NH_4)_2SO_4/H_2SO_4$ seed particles the oligomer fraction increases with increasing

RH (Fig. 5b). As it was mentioned, aldol condensation can be assumed as the favored accretion reaction under
these conditions (Yasmeen et al., 2010). Aldol condensation includes as a first step aldol addition followed by
a loss of water. The loss of water is irreversible, thus the aldol condensation will not be inhibited with higher
LWC of the seed particles.

Other accretion reactions can contribute to the formation of heat-decomposable methylglyoxal

oligomers with $(NH_4)_2SO_4/H_2SO_4$ seed particles as well. Altieri et al. (2008) detected products formed through
acid-catalysed esterification at pH $\approx$ 4. This equilibrium reaction involves the reversible loss of water as it was
reported for acetal/hemiacetal formation (Lim et al., 2010). Thus, it can be expected that with higher LWCs
the contribution of esterification reactions to oligomer formation decreases due to the shift of the equilibrium
towards the monomers.

Imidazole formation was also postulated as possible oligomer-formation mechanism for

methylglyoxal (Sedehi et al., 2013; De Haan et al., 2011). It was found that imidazole formation is of minor
importance compared to aldol condensation (Sedehi et al., 2013). However, imidazole formation involves also
a reversible loss of water, thus it does not provide a feasible explanation for the higher oligomer fraction at
higher RH with $(NH_4)_2SO_4/H_2SO_4$ seed particles.

Radical – radical reactions are also postulated as a possible reaction pathway to form oligomers

(Schaefer et al., 2015; Lim et al., 2013; Rincon et al., 2009; Lim et al., 2010; Sun et al., 2010). Radical-radical
reactions of methylglyoxal might occur following the H - atom abstraction of methylglyoxal with OH radicals
and a subsequent recombination of the resulting alkyl radicals (as discussed for glyoxal in Schaefer et al.,
2015). The contribution of radical – radical reactions to oligomer formation is not well understood as
obviously, the reaction of alkyl radicals with oxygen tends to suppress this pathway. Nevertheless, it can be
expected, that with higher LWCs of the seed particles and thus with a higher reaction volume, the absolute
amount of methylglyoxal in the particle phase might increase, but not its particle-phase concentrations. For
aerosol particle systems, ionic strength effects (Herrmann et al., 2015) are able to influence the uptake of
methylglyoxal into the particle phase as well. Waxman et al. (2015) observed a salting-out effect for
methylglyoxal for all investigated seed particles at higher ionic strengths.

Low pH combined with high solute concentrations as calculated for the present aerosol particles, can

trigger isomerisation (or switching) reactions as discussed by Herrmann et al. (2015). Overall, a clear
discussion on how radical-radical reaction might be affected by increasing LWC (through increasing RH) and
by pH is difficult at the current level of knowledge.

In summary, the present study provides a reliable quantification method for heat-decomposable

methylglyoxal oligomers formed by 1,3,5-TMB oxidation. The fraction of oligomeric substances formed solely
by methylglyoxal oligomerisation varied dependent on RH and seed particle acidity between 2 – 8%, which is
lower than the determined values by Kalberer et al. (2004) and Baltensperger et al. (2005) (varying between
50 and 80%). In the present study only heat-decomposable methylglyoxal oligomers were quantified, thus
there might be not heat-decomposable methylglyoxal oligomers or oligomers originating from other monomers
than methylglyoxal, which were not determined in the present study leading to lower oligomer fractions of $\Delta M$
compared to the literature studies. The obtained data are not fully conclusive and literature studies are often
contradicting. Thus, more experiments are necessary to get a clearer picture about the influence of RH and
particle phase acidity on oligomer formation and to explain the non-linear relation between the oligomer
fraction and RH.

## 4. Summary

In the present study a method was developed to quantify oligomers formed from methylglyoxal. The method is based on the thermal decomposition of heat-decomposable methylglyoxal oligomers into monomers. The formed methylglyoxal monomers were detected with PFBHA derivatisation and GC/MS analysis. The influence of heating time, pH and heating temperature on the decomposition of heat-decomposable methylglyoxal oligomers was systematically investigated. The best result was achieved with a heating time of 24 hours at 100°C and pH = 1. The method was applied to heat-decomposable methylglyoxal oligomers formed during the oxidation of 1,3,5-TMB resulting in an oligomer fraction of up to $\approx$ 8%. A contradicting dependency of the oligomer fraction under varying RH with $NH_4HSO_4$ and $(NH_4)_2SO_4/H_2SO_4$ seed particles was found, which might be caused by different oligomer formation mechanisms.

Overall, the present method provides an important step revealing the amount of oligomers present in the particle phase, their tentative formation mechanism and their importance for aqSOA formation.

## 5. Acknowledgements

This study was supported by the Scholarship program of the German Federal Environmental Foundation (Deutsche Bundesstiftung Umwelt, DBU; grant number 20013/244).

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

Table 1: Overview of methods for structure elucidation of oligomeric compounds (after Hallquist et al., 2009).

| Method | Reference |
|---|---|
| (Matrix assisted) laser desorption/ionisation mass spectrometry (MALDI-MS, LDI-MS) | Dommen et al., 2006; Kalberer et al., 2004; Kalberer et al., 2006; Reinhardt et al., 2007; Holmes and Petrucci, 2006; Surratt et al., 2006; Denkenberger et al., 2007 |
| Fourier transform ion cyclotron resonance mass spectrometry (FT-ICR-MS) | Kundu et al., 2012; Altieri et al., 2008; Tolocka et al., 2004; Hall and Johnston, 2012; Denkenberger et al., 2007; Tan et al., 2012 |
| On-line atmospheric pressure chemical ionisation tandem mass spectrometry (APCI tandem MS) | Müller et al., 2008 |
| Aerosol mass spectrometry (AMS) | Sareen et al., 2010; Schwier et al., 2010; Bahreini et al., 2005; Heaton et al., 2007 |
| Electrospray ionisation mass spectrometry (ESI/MS, ESI/MS/MS) | Altieri et al., 2008; Hall and Johnston, 2012; Surratt et al., 2006; Yasmeen et al., 2010; Hastings et al., 2005; Bones et al., 2010; Surratt et al., 2007; Hamilton et al., 2006; Sadezky et al., 2006; Sato et al., 2012; Noziere et al., 2010; Tolocka et al., 2004; Iinuma et al., 2004; Nguyen et al., 2011; Bahreini et al., 2005 |
| Aerosol time of flight mass spectrometry (ALTOFMS) | Huang et al., 2015 |
| Gas chromatography mass spectrometry (GC/MS) | Hastings et al., 2005; Surratt et al., 2006; Szmigielski et al., 2007; Angove et al., 2006 |
| Ion trap mass spectrometry (IT-MS) | Surratt et al., 2006; Gao et al., 2004 |
| Photoelectron resonance capture ionisation-aerosol mass spectrometry (PERCI-MS) | Zahardis et al., 2005 |
| Ultraviolet-visible spectroscopy (UV/Vis) | Nemet et al., 2004; Noziere and Esteve, 2005; Bones et al., 2010; Song et al., 2013; Casale et al., 2007; Alfarra et al., 2006; Drozd and McNeill, 2014; Noziere and Cordova, 2008 |
| Fourier transform infrared spectroscopy (FTIR) | Loeffler et al., 2006; Bones et al., 2010; Jang et al., 2003; Jang and Kamens, 2001; Holmes and Petrucci, 2006 |
| nuclear magnetic resonance spectroscopy (NMR) | Nemet et al., 2004; Bones et al., 2010; Angove et al., 2006; Garland et al., 2006; Kua et al., 2013; De Haan et al., 2011 |

Table 2: Experiments in the aerosol chamber LEAK for the OH radical oxidation of 1,3,5-TMB. All experiments were conducted at 293 K and with 91.9 ppb 1,3,5-TMB.

| Experiment number | seed | RH [%] | $O_{3ini}$ [ppb] | $\Delta HC$ [ppb] | $\Delta M$ [µg m$^{-3}$] | SOA yield $Y_{SOA}$ [%] |
|---|---|---|---|---|---|---|
| #1 | 78 mmol L$^{-1}$ NH$_4$HSO$_4$ | $\approx 0$ | $\approx 137$ | 57.2 | 19.7 | 7.00 |
| #2 | 78 mmol L$^{-1}$ NH$_4$HSO$_4$ | 50 | $\approx 133$ | 55.2 | 11.3 | 4.12 |
| #3 | 78 mmol L$^{-1}$ NH$_4$HSO$_4$ | 75 | $\approx 134$ | 56.2 | 14.2 | 5.14 |
| #4 | 60 mmol L$^{-1}$ (NH$_4$)$_2$SO$_4$/ 0.4 mmol L$^{-1}$ H$_2$SO$_4$ | $\approx 0$ | $\approx 132$ | 56.2 | 18.1 | 6.55 |
| #5 | 60 mmol L$^{-1}$ (NH$_4$)$_2$SO$_4$/ 0.4 mmol L$^{-1}$ H$_2$SO$_4$ | 50 | $\approx 135$ | 56.5 | 11.7 | 4.21 |
| #6 | 60 mmol L$^{-1}$ (NH$_4$)$_2$SO$_4$/ 0.4 mmol L$^{-1}$ H$_2$SO$_4$ | 75 | $\approx 144$ | 57.2 | 13.9 | 4.94 |

1,3,5-TMB: 1,3,5-Trimethylbenzene; RH: Relative humidity

Table 3: Investigated method parameters and used PTFE filters from 1,3,5-TMB oxidation for method development.

| Parameter | Range | Filter of experiment |
|---|---|---|
| Heating time | 15, **24**, 30, 48 hours | #3 |
| pH | **1**, 3, 5, 7 | #2 |
| Heating temperature | 50, **100** °C | #2 |

Selected parameters given in **bold**.

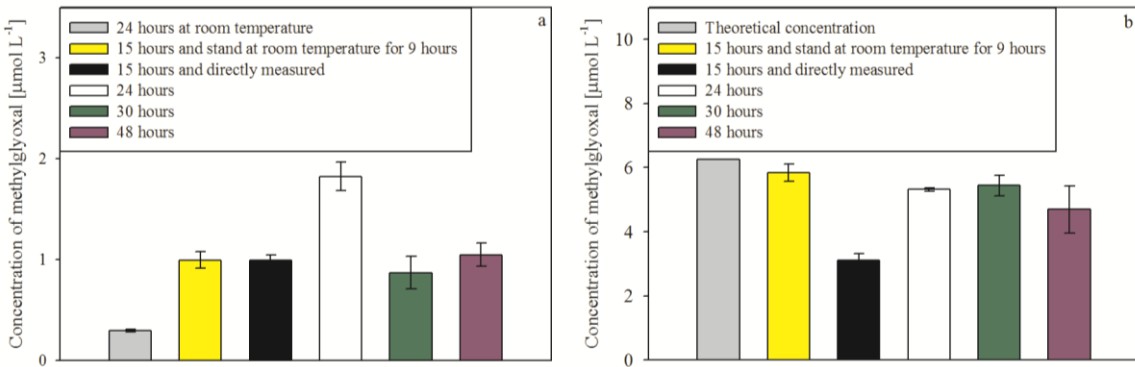

Figure 1: Influence of the heating time on the detected methylglyoxal concentrations in filter samples (a) and the methylglyoxal standard solution (b).

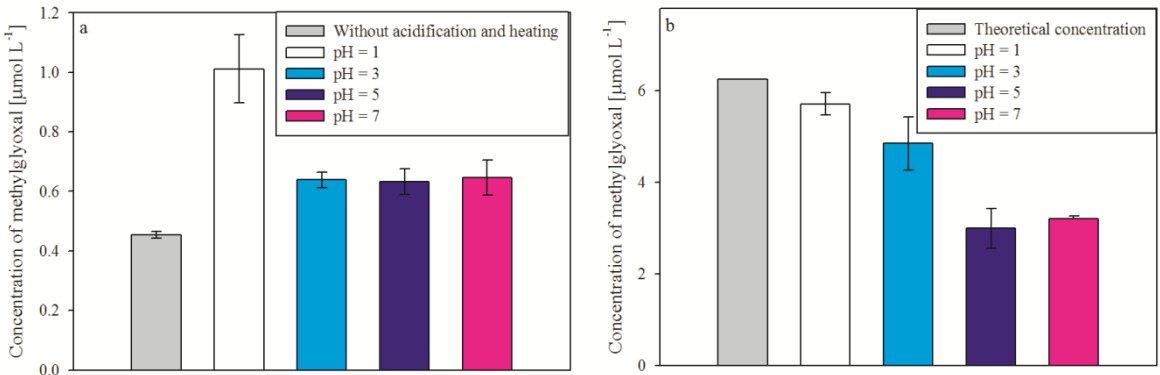

Figure 2: Influence of pH on the detected methylglyoxal concentration in filter samples (a) and the methylglyoxal standard solution (b).

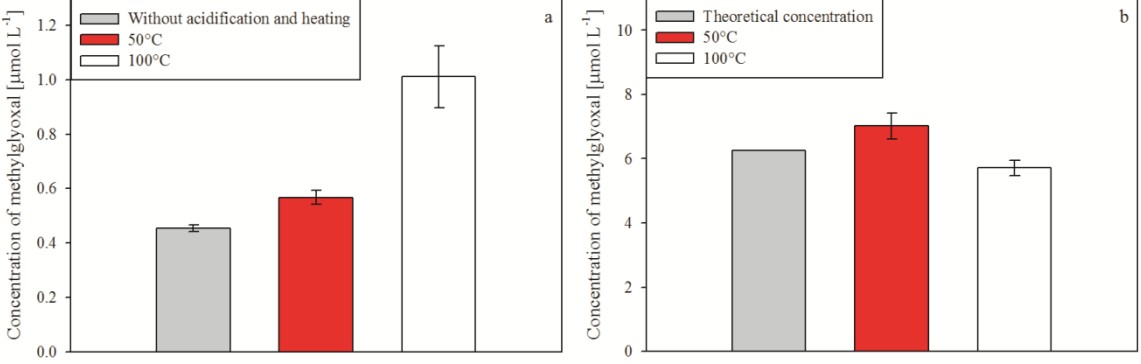

Figure 3: Influence of the heating temperature on the detected methylglyoxal concentration in filter samples (a) and the standard solution (b).

Table 4: SOA yields ($Y_{SOA}$) of 1,3,5-TMB reported in the literature.

| SOA yield $Y_{SOA}$ [%] | Reference |
|---|---|
| 2.81 - 7.91 | Cocker et al., 2001 |
| 4.5 – 8.34 | Healy et al., 2008 |
| 2.5 ± 0.1 – 15.6 ± 1.0 | Sato et al., 2012 |
| 3.1 | Odum et al., 1997 |
| 0.41 ± 0.1 | Kleindienst et al., 1999 |
| 0.29 – 6.36 | Wyche et al., 2009 |
| 4.7 ± 0.7 | Paulsen et al., 2005 |
| 7.1 ± 0.3 – 13.8 ± 0.6 | Cao and Jang, 2007 |
| 4.12 – 7.00 | This work |

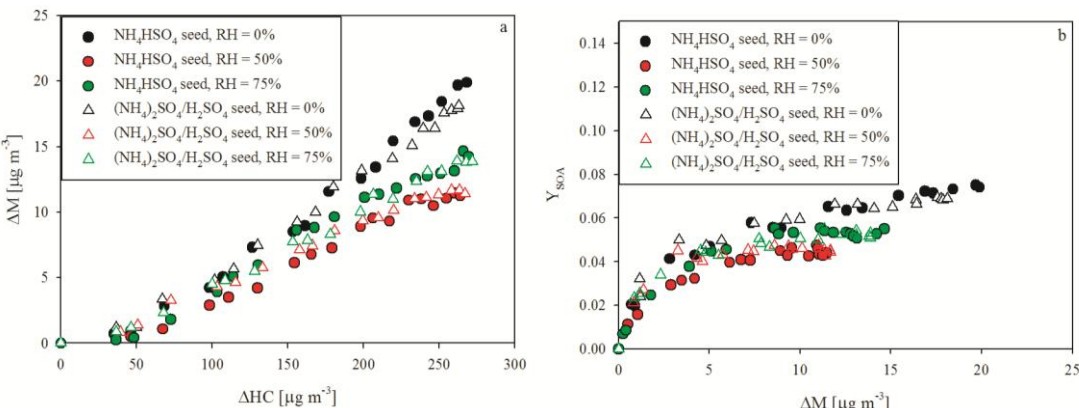

Figure 4: SOA growth curve (a) and yield curve (b) of the OH radical oxidation of 1,3,5-TMB in the presence of $NH_4HSO_4$ or $(NH_4)_2SO_4/H_2SO_4$ seed under variation of RH.

Table 5: LWC and pH of the seed particles calculated with E-AIM.

| seed | RH [%] | pH$_{seed}$[a] | LWC [g m$^{-3}$][a] |
|---|---|---|---|
| 78 mmol L$^{-1}$ $NH_4HSO_4$ | ≈ 0 | _[b] | _[b] |
| 78 mmol L$^{-1}$ $NH_4HSO_4$ | 50 | 0.1 | $6.66 \times 10^{-6}$ |
| 78 mmol L$^{-1}$ $NH_4HSO_4$ | 75 | 1.2 | $12.29 \times 10^{-6}$ |
| 60 mmol L$^{-1}$ $(NH_4)_2SO_4$/0.4 mmol L$^{-1}$ $H_2SO_4$ | ≈ 0 | _[b] | _[b] |
| 60 mmol L$^{-1}$ $(NH_4)_2SO_4$/0.4 mmol L$^{-1}$ $H_2SO_4$ | 50 | 4.0 | $4.25 \times 10^{-6}$ |
| 60 mmol L$^{-1}$ $(NH_4)_2SO_4$/0.4 mmol L$^{-1}$ $H_2SO_4$ | 75 | 4.2 | $11.56 \times 10^{-6}$ |

LWC: Liquid water content; [a] pH and LWC of the seed particles were calculated for different RH using model II from the extended aerosol thermodynamic model (E-AIM; Clegg et al., 1998); [b] calculation of the pH and LWC was not possible due to the low relative humidity of RH ≈ 0%. A RH = 10% is set as lower limit in E-AIM.

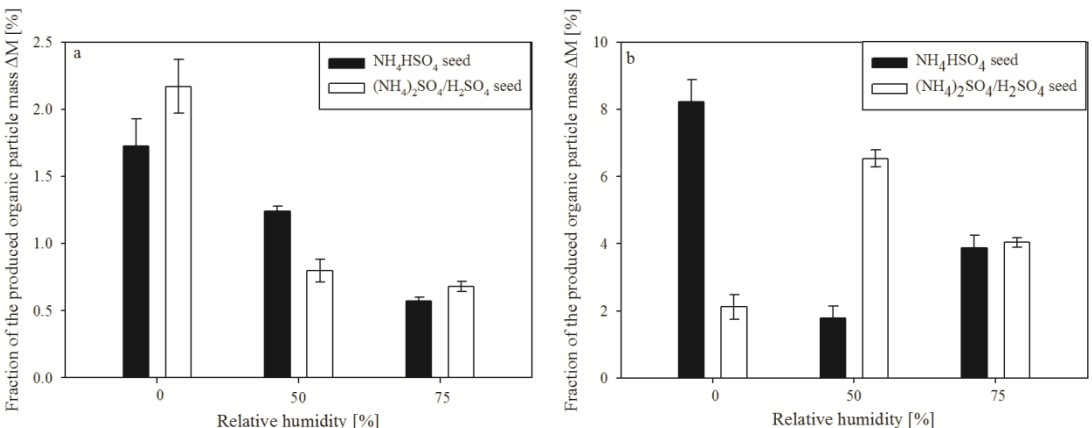

Figure 5: Contribution of methylglyoxal (a) and their heat-decomposable oligomers (b) to the produced organic particle mass (ΔM) with $NH_4HSO_4$ and $(NH_4)_2SO_4/H_2SO_4$ seed particles under variation of the relative humidity.