# Peer review of "A quantification method for heat-decomposable methylglyoxal oligomers and its application on 1,3,5trimethylbenzene SOA"

_Atmospheric Chemistry and Physics, 2016_

## Referee Comment (RC1) · Anonymous Referee #1 · 24 Aug 2016

General Comments

This manuscript describes the careful optimization of a method designed to thermally break down methylglyoxal oligomers and detect them as derivatized monomers. Such a method is a welcome addition to the experimental toolbox of aerosol chemists. Overall, the methods and data are clearly explained and the conclusions are reasoned out logically. While the optimization process and results are completely valid, the authors' ability to interpret their work is limited by the fact that standards for methylglyoxal oligomers do not exist. To be more precise, the authors have not demonstrated that the maximum amount of oligomers that they have been able to break down and detect is equal to the total amount of methylgyloxal oligomers in their samples. However, the

manuscript seems to assert this questionable assumption many times. In fact, as the authors point out at the end, the 5 – 10-fold difference between methylglyoxal oligomers detected in this study and total oligomers detected by different methods in other studies of the same system suggests that many other oligomers are present, but don't either don't break down at all, or don't break down into monomers that were detectable using the present method. However, the lack of detection of any other carbonyl compounds mentioned in this study suggests to this reviewer that the undetected oligomers are most likely some type of methylglyoxal oligomer that is irreversibly formed. It may be that methylglyoxal acetal oligomers break down to monomers with heat (as is the case for glyoxal), but methylglyoxal aldol oligomers do not. Until there are methylglyoxal oligomer standards of each type to test, this question is not likely to be answered definitively. To address this issue, the authors should qualify all of their statements (including even the title) about what they are detecting, perhaps by replacing "methylglyoxal oligomers" with a more exact phrase such as "reversibly-formed methylglyoxal oligomers."

This work will be of interest to those who study the chemistry of organic aerosol particles in the lab and in the field.

Specific Comments

Line 19: Some qualifying phrase seems to be missing here. Is this percentage the fraction of anthropogenic emissions, or of non-methane hydrocarbon emissions? Are aromatic emissions in China really 52% of the total of all biogenic and anthropogenic hydrocarbon emissions, including methane?

Line 92: It would be helpful to readers if a 1-sentence summary could be included here about how O3 + TME generates OH radicals and why this OH source was chosen.

Line 96: Are the deliquescence and efflorescence RH values known for either seed particle material? Perhaps the phase of these particles is relevant to the results.

Line 99: Is the choice of organic aerosol density (1 g/mL) based on a literature measurement?

Line 102: What absorbent material were the denuders coated with?

Line 191: There is unclear logic here. If the incompletely derivatized methylglyoxal is able to reform oligomers once the temperature drops to 298 at t = 15 h, wouldn't the yellow bar in Figure 1b be even smaller than the black bar?

Line 195: This paragraph is confusing. The authors suggest a possible explanation, then seem to eliminate it in the next sentence (assuming that "decomposition" and "loss" are the same process), without offering an alternative explanation.

Line 215: The manuscript states here that Rodigast et al. 2015 found no influence of the pH on the PFBHA derivatisation reaction. However, in line 201 the same study is cited in support of an optimal pH value of 1 for the reaction. How can a factor with no influence be optimized?

Line 232: This is an important statement, but it leaves the reader wondering which other carbonyl compounds were being monitored in this work. It would be very helpful if the authors would list these compounds.

Line 236: Confusing statement. According to Table 3, the method was developed using filters from experiments 2 and 3 from laboratory 1,3,5-TMB oxidation studies (plus methylglyoxal monomer standards). Now the method is applied to more of the samples from the same set of studies.

Line 257: From the information given it is hard to see how these two studies "are in good agreement" with the present study, since the present study is measuring no acidity effect while the two literature studies compared seed / no seed conditions.

Line 289: The authors' measurements of the percentage methylglyoxal in 1,3,5-TMB SOA are right in line with other published values. Did Healy et al and Cocker et al. use seed particles? Can the authors comment on whether such percentages are consistent

with partitioning theory and the vapor pressure of methylglyoxal?

Line 301: Do these literature studies suggest that either of the two explanations offered in the previous sentence are more likely than the other?

Line 315: The phrase "opposite trend" was unclear. What trend is this trend the opposite of?

Line 334: Doesn't this logic also apply to the next two reactions that involve water loss? I am unsure why aldol reactions are enhanced and esterification and imidazole formation are decreased at high RH, when all involve water loss as part of the mechanism.

Technical Corrections

Line 194: "might be" should be "are". The authors know that H+ and NH4+ ions are present in the samples and therefore the aqueous extracts.

Line 205: "PFTFE" should be "PTFE"

Line 253: "condensate" should be "condense"

Line 314: The period is missing after "conditions"

Line 343: "is" should be "it"

Line 352: "increases" should be "increase"

Table 6 does not provide any additional information beyond Figure 5b, and could be eliminated.

---

## Referee Comment (RC2) · Anonymous Referee #2 · 11 Nov 2016

Overall Comment and Recommendation:

This study is primarily a method development study aimed at quantifying oligomers derived from multiphase chemistry of methylglyoxal by using GC/MS with prior derivatization. Many groups have shown that when you heat certain SOA types, such as IEPOX-derived SOA, you measure monomeric products (like 2-methyltetrols) in high quantities (Lopez-Hilfiker et al., 2016, ES&T). This is important since monomeric products like 2-methyltetrols are too volatile to exist in such large quantities to explain the observed SOA formation in lab or field studies. The present study utilized GC/MS with PFBHA derivatization to detect methylglyoxal monomers found in 1,3,5-TMB derived SOA. The authors systematically examined the influence of heating time, pH, and heating temperature on the decomposition of methylglyoxal oligomers. The authors found that the best result was likely acheived when heating the extracts for 24 hours at 100 degrees C and at pH of 1. The authors found that the oligomers accounted for up to 8% of the total SOA mass.

The method the authors develop could be very useful in trying to provide mass closure of the oligomer fraction of methylglyoxal-derived oligomers. More importantly, this method could likely be adapted to determine oligomer mass fractions in other types of SOA. This is important since we currently lack appropriate authentic standards to quantify the oligomer content of the SOA. More often we have standards available to quantify the monomers, so this method would be of interest to many research groups working in this area. Before I can recommend final publicaiton in ACP, I have many specific comments that need to be addressed by the authors. These specific comments are outlined below. Due to the nature of these comments, I must recommend to the Editor that this manuscript be accepted with major revisions noted. There are several method details missing that need to be clarified or added to the main text. In addition, in several sections of the manuscript, the English writing is at times quite poor. As a result, I encourage the authors to conduct further editing on the writing before resubmission.

Specific Comments:

1.) Citations in main text:

The authors cite references through the manuscript using "Last name of first author et al., Year." As an example, please refer to Page 2, LIne 52. The the authors should change to Kalberer et al. (2004). The style in ACP is always "Author last name et al. (year)."

2.) Filter Extractions:

Have the authors tested extracting the filters in an organic solvent such as acetontrile

or methanol? I wonder how the extraction efficiencies of potentially large oligomers change with extraction solvent? This factor should at least be discussed in this manuscript and the likely uncertainties in obtaining exact oligomer concentrations from this method. What I'm getting at is the authors assume in the text that the filter extraction efficiency is likely 100% in water.

3.) GC/MS operating details:

You should state here explicitly how long your GC/MS run is. Since it appears this is a long GC/MS run, did the authors check to see how the methylglyoxal standard calibration changes throughout the run? For example, did the authors consider rerunning the calibration at the end of the run? Did the response factor change/drift dramatically?

4.) Method Development:

In the method development section when varying heating time, pH, and heating temperature, it seems that for the latter two the same experiment was used (i.e., experiment 2). In contrast, the heating time was explored with experiment 3. I think the authors would have been better to use the same type of aerosol generated under the exact same conditions when exploring these parameters of the method. I'm curious to know why this wasn't done or why it is justified to do this as is?

5.) SOA Yields:

Since the authors spend time in this study reporting SOA yields from 1,3,5-TMB oxidation, I have some questions about the differences in the amount of organic aerosol produced under the different RH conditions. Considering that you observe more SOA under dry conditions when compared to more humid conditions, I wonder what role your chamber walls are playing? Recent work from the Caltech (Seinfeld), CU-Boulder (Jimenez and Ziemann), and CMU (Donahue) groups suggest that the wall effect could be really important, especially if your goal is to report SOA yields in the literature. Thus,

maybe you are losing more things to your wall under higher RH conditions? Have the authors considered how to correct SOA yields for this effect? If not, you should at least acknowledge the likely importance of wall losses of semivolatile and less volatile organic vapors.

6.) Page 9, Lines 298-299:

Could it be that methylglyoxal's chamber wall losses are also changing with RH? Did the authors consider injecting methylgloxal in the gas phase of the chamber and investigate its wall losses with different RHs? That might provide more insights into the importance of your chamber wall.

7.) Page 9, Lines 329-332:

I wonder how the different seed aerosols you use might cause differences in aerosol phase separation/morphology? What role could this potentially have in explaining the differences in the oligomer fraction?

8.) Oligomer Types:

It appears that the authors only consider the oligomer type resulting from methylglyoxal + methylglyoxal type reactions. However, considering the plethora of other monomers when oxidizing a VOC like 1,3,5-TMB, why did the authors not consider other types of oligomer reactions involving methylgloyxal + some other oxidized product? Was there no evidence for this in your GC/MS data? Related to this, why didn't the authors provide a TIC or EIC in the main text? In either the TIC or EIC, it would be helpful to provide peak labels and likely respective mass spectra to each chromatographic peak.

I mention this as recent work by Lin et al. (2014, ES&T) demonstrated the varying types of IEPOX-derived oligomers under different RH and seed aerosol conditions. It appeared from their LC/MS data that there was a very wide degree of types (e.g., light- versus non-light absorbing) and lengths of oligomers present.

Minor Comments:

1.) Introduction, Lines 36-42: When generalizing the oligomeric mechanisms leading to SOA, why not include those derived from acid-catalyzed hydrolysis of epoxides (e.g., Paulot et al., 2009, Science, Surratt et al., 2010, PNAS; Lin et al., 2014, ES&T)?

2.) Page 2, Line 46. You should probably define the ESI/MS/MS acronym being associated with tandem mass spectrometry interfaced to ESI.

3.) Page 2, Line 50:

Probably change "the effort for structure" to "past efforts for structural"

4.) Page 3, Line 72:

Do you mean to say "fundamental" here instead of "foundation"?

5.) Page 3, Line 87:

should this say instead: "was investigated in the Leipziger AerosolKammer (LEAK) chamber"?

6.) Page 3, Line 90:

Maybe change "ammonium hydrogensulfate" to "ammonium bisulfate?

7.) Page 3, Line 90:

Maybe change "ammoniumsulfate" to "ammonium sulfate"?

8.) Page 4, Line 92:

Add vendor and model to "(PTR-TOF MS)"

9.) Page 4, Line 99:

Add vendor and model to "(SMPS)"

10.) Page 4, Line 99:

Why do the authors use 1 g cm-3 density? Are you using this based on a previous

study? If so, please justify why you used this aerosol density.

11.) Page 4, Line 100:

Change "the particle phase" to "aerosol"

12.) Page 4, Line 100:

Insert comma between "experiments" and "1.2"

13.) Page 4, Line 104:

Did the authors determine what the break through could be on these filters during experiments? Were control tests done to know how well the denuder worked?

14.) Page 7, Line 252:

Change "condensate" to "condense"

15.) Page 8, Lines 279-281:

Citation is needed here. Are the authors arguing that particle-phase acidity might also be required for methylglyoxal oligomers to form? If so, is this why you think LWC matters? That is, the higher the LWC the more likely the aerosol pH is less acidic and thus affecting the amount of SOA due to oligomer formation? This is unclear to me in the current text.

16.) Page 9, Line 308:

Do the authors mean to say "on average $\sim$ 2%"?

17.) Page 10, Line 350:

Change "increases" to "increase"

---

## Author Comment (AC1) · 5 Jan 2017

Responses to Reviewers' Comments:

We thank the reviewers for their constructive comments. The manuscript is revised based on the suggestions made and detailed responses to the reviewers are addressed as follows.

**Referee #1**

This manuscript describes the careful optimization of a method designed to thermally break down methylglyoxal oligomers and detect them as derivatized monomers. Such a method is a welcome addition to the experimental toolbox of aerosol chemists. Overall, the methods and data are clearly explained and the conclusions are reasoned out logically. While the optimization process and results are completely valid, the authors' ability to interpret their work is limited by the fact that standards for methylglyoxal oligomers do not exist. To be more precise, the authors have not demonstrated that the maximum amount of oligomers that they have been able to break down and detect is equal to the total amount of methylgyloxal oligomers in their samples. However, the manuscript seems to assert this questionable assumption many times. In fact, as the authors point out at the end, the 5 – 10-fold difference between methylglyoxal oligomers detected in this study and total oligomers detected by different methods in other studies of the same system suggests that many other oligomers are present, but don't either don't break down at all, or don't break down into monomers that were detectable using the present method. However, the lack of detection of any other carbonyl compounds mentioned in this study suggests to this reviewer that the undetected oligomers are most likely some type of methylglyoxal oligomer that is irreversibly formed. It may be that methylglyoxal acetal oligomers break down to monomers with heat (as is the case for glyoxal), but methylglyoxal aldol oligomers do not. Until there are methylglyoxal oligomer standards of each type to test, this question is not likely to be answered definitively.

1) To address this issue, the authors should qualify all of their statements (including even the title) about what they are detecting, perhaps by replacing "methylglyoxal oligomers" with a more exact phrase such as "reversibly-formed methylglyoxal oligomers."

Author`s comment

The authors agree with the reviewer. Nevertheless, it is not clear if the quantified oligomeric compounds are formed reversible. It is also possible to form oligomers for example due to aldol condensation, which is an irreversible oligomerisation reaction. Thus, the term "reversibly-formed methylglyoxal oligomers" cannot be easily applied.

In the present study it is assumed that the heating process decomposes methylglyoxal oligomers into monomers. Therefore, only those oligomeric compounds are detected which can be decomposed through the heat treatment which are now addressed as "heat-decomposable".

Thus, the term *"(methylglyoxal) oligomers"* was changed to *"heat-decomposable (methylglyoxal) oligomers"* in the manuscript and the paragraph (Page 3, Line 72) *"Thus, the present study presents a fundamental approach for a reliable quantification of methylglyoxal oligomers in laboratory-generated SOA. The method is applicable for all oligomeric compounds, which can be decomposed into methylglyoxal monomers during the heating process at a temperature of 100 °C. As the oligomerisation mechanisms leading to the quantified oligomeric compounds are not known, it cannot be specified if the oligomers are reversibly or irreversibly formed. In addition, it cannot be excluded that there exist oligomers, which are not decomposable into their methylglyoxal monomers through the heating process. Thus, the quantified oligomers are termed as heat-decomposable methylglyoxal oligomers."* was included in the manuscript.

This work will be of interest to those who study the chemistry of organic aerosol particles in the lab and in the field.

Specific Comments

2) Line 19:  Some qualifying phrase seems to be missing here. Is this percentage the fraction of anthropogenic emissions, or of non-methane hydrocarbon emissions?  Are aromatic emissions in China really 52% of the total of all biogenic and anthropogenic hydrocarbon emissions, including methane?

Author`s comment

The percentage is related to the emission of non-methane hydrocarbons measured at an industrial location. The emission is dominated by, e.g., textile, shoe and furniture manufactures. In the same study it is mentioned that at areas, which are not near to an industrial location, alkanes have the highest contribution to non-methane hydrocarbon emission in china.

The sentence (Page 2, Line 19) was changed to *"[...] with up to 52% to the total non-methane hydrocarbon mass at an industrial dominated site in China (Liu et al., 2008)."*

3) Line 92: It would be helpful to readers if a 1-sentence summary could be included here about how $O_3$ + TME generates OH radicals and why this OH source was chosen.

Author`s comment

The ozonolysis of TME was chosen as OH-radical source because it generates OH radicals in the dark and under low $NO_x$ conditions. It was found in previous experiments that the oxidation of 1,3,5-TMB at high $NO_x$ levels, e.g., with HONO as OH-radical source leads to a negligible particle growth ($\Delta HC = 23$ ppb; $\Delta M = 0$ µg m$^{-3}$). Thus, the investigation of SOA formation and of particle-phase products like oligomers is not possible under these conditions. Additionally, the photolysis of $H_2O_2$ was used as OH-radical source (low $NO_x$) but the applied UV-C light ($\lambda = 254$ nm) results in photolysis of the aromatic precursor compound. Thus, the photolysis of $H_2O_2$ is also not suitable as OH-radical source for the oxidation of 1,3,5-TMB. Consequently, SOA originated from 1,3,5-TMB oxidation has to be examined at low $NO_x$ levels and without UV-C light. For that reason, the ozonolysis of TME was used, which is an OH-radical source under dark and low $NO_x$ conditions.

The sentence (Page 3, Line 91) was changed to *"In order to investigate OH-radical oxidation of 1,3,5-TMB at low $NO_x$ levels (< 1 ppb) and under dark conditions the ozonolysis of tetramethylethylene (TME) was used as OH-radical source (Berndt and Böge, 2006)."*

The paragraph (Page 3, Line 92) *"The cycloaddition of ozone to TME yields a primary ozonide, which reacts further and forms a stabilised Criegee Intermediate (sCI). The sCI decomposes via the hydroperoxide channel, leading to the formation of OH radicals (Gutbrod et al., 1996) with a yield of 0.92 ± 0.08 (Berndt and Böge, 2006)."* was added.

4) Line 96: Are the deliquescence and efflorescence RH values known for either seed particle material? Perhaps the phase of these particles is relevant to the results.

Author´s comment

In general and as an simplified approximation, atmospheric particles can be solid or liquid (Ziemann, 2010). The phase state can be changed due to deliquescence and efflorescence of the particles. The phase of the particles might have an influence on the partitioning between the gas and the particle phase, reactions in the particle phase, the mass transport of the reactants and oxidants into the particle phase, and the water uptake of the particles (Ziemann, 2010; Saukko et al., 2012).

For pure $NH_4HSO_4$ and $(NH_4)_2SO_4$ seed particles the deliquescence RH and efflorescence RH can be found in the literature. The deliquescence RH of $(NH_4)_2SO_4$ and $NH_4HSO_4$ seed particles were reported in the literature to be 79% and 39%, respectively (Cziczo et al., 1997). Thus, pure $NH_4HSO_4$ seed particles are liquid at the applied RH values of 50% and 75%. The efflorescence RH was measured in the literature as 33% for $(NH_4)_2SO_4$ seed particles and lower than 2% for

$NH_4HSO_4$ seed particles (Cziczo et al., 1997; Mikhailov et al., 2009). Thus, pure $(NH_4)_2SO_4$ seed particles might be solid over the whole RH range applied in the present study while pure $NH_4HSO_4$ particles are solid only at RH = 0%.

Nevertheless, for mixed particles including organic and inorganic compounds (as in the present study) the knowledge about the exact phase state is limited. The organic compounds change the deliquescence point (Andrews and Larson, 1993; Lightstone et al., 2000) as well as the hygroscopic behaviour of the particles (Lightstone et al., 2000; Prenni et al., 2003; Chen and Lee, 1999). This is strongly dependent on the type of the organic compounds and their properties. As the composition of the organic phase in the particles is not comprehensively elucidated, the phase state of the particles in the present study is not known.

However, Virtanen et al., 2010 postulated that particles are in an amorphous solid state if oligomeric compounds are present in the particles. Thus, it can be speculated that the particles in the present study containing a fraction of up to 8% of oligomeric compounds are also in an amorphous solid phase state changing with the fraction of oligomers and other organic compounds. Thus, the particles can have different phases, which might influence the product distribution in the particle phase and therefore the SOA yields.

The following paragraph was included in the manuscript (Page 8, Line 259): *"The RH value can have an influence on the phase state of the particles while the phase state has an effect on the partitioning of the compounds into the particles and the particle-phase reactions (Ziemann, 2010; Saukko et al., 2012). In the present study the particles are a mixture of inorganic and partitioned organic compounds, thus the phase state is not known but it might be possible that the phase state influences ΔM and the SOA yields."*

5) Line 99: Is the choice of organic aerosol density (1 g/mL) based on a literature measurement?

Author´s comment

Different values can be found in the literature. Most of the studies assumed a density of 1.4 g cm$^{-3}$ (Sato et al., 2012; Praplan et al., 2014; Müller et al., 2012) based on the measurements by Alfarra et al., 2006, which determined values between 1.35 g cm$^{-3}$ and 1.40 g cm$^{-3}$. Kleindienst et al., 1999 used a density of 1 g cm$^{-3}$, which was used in the present study as well.

6) Line 102: What absorbent material were the denuders coated with?

Author`s comment

The denuders are coated with XAD-4, which enables the trap of gas-phase compounds and avoid artefacts of gas-phase products on the PTFE-filters (Kahnt et al., 2011).

The sentence (Page 4, Line 100) was changed to *"[…] experiments, 1.2 m³ of the chamber volume was sampled on a PTFE filter (borosilicate glass fiber filter coated with fluorocarbon, 47 mm in diameter, PALLFLEX T60A20, PALL, NY, USA) connected to a XAD-4 coated denuder […]."*

7) Line 191: There is unclear logic here. If the incompletely derivatized methylglyoxal is able to reform oligomers once the temperature drops to 298 at t = 15 h, wouldn't the yellow bar in Figure 1b be even smaller than the black bar?

Author´s comment

The authors agree with the reviewer and include a paragraph to clarify the results.

The authors assumed that the derivatisation as well as the oligomerisation proceeds during the 9 hours at room temperature (sample a). In comparison, sample b was directly measured after 15 hours heating thus, there is no further derivatisation and oligomerisation. Altogether, the derivisation time of sample a (yellow bar) was longer than of sample b (black bar) leading to higher methylglyoxal concentrations even if oligomerisation occurs in sample a as well.

The sentence *"Despite the oligomerisation of methylglyoxal monomers during the 9 hours at room temperature, the derivatisation proceeds as well during this time leading to higher methylglyoxal concentrations in sample a than in sample b, which was directly measured after 15 hours heating (Fig. 1b)."* was added (Page 6, Line 195).

In addition, the sentence (Page 6; Line 185) is misleading and was changed to *"To probe this hypothesis a 6.25 µmol L$^{-1}$ methylglyoxal standard was heated for 15 hours and measured immediately (like sample b) or, alternatively, was allowed to stand at room temperature for 9 hours (like sample a)."*

8) Line 195: This paragraph is confusing. The authors suggest a possible explanation, then seem to eliminate it in the next sentence (assuming that "decomposition" and "loss" are the same process), without offering an alternative explanation.

Author´s comment

The authors agree with the reviewer but this cannot be conclusively clarified. It can only be mentioned that no influence on the derivatisation reaction was found (Fig. 1b).

It can be suggested that there are additional reactions in the filter samples perhaps due to other particle-phase species, which do not exist in the standard sample. These further reactions might lead to lower concentrations in the filter samples after a heating time of 24 hours, which was not observed for the methylglyoxal standard solution.

The sentence (Page 6, Line 200) *"Thus, it can be speculated that the low methylglyoxal concentrations in the filter samples are a result of further reactions with particle-phase species, which do not exist in the standard samples."* was included in the manuscript.

9) Line 215: The manuscript states here that Rodigast et al. 2015 found no influence of the pH on the PFBHA derivatisation reaction. However, in line 201 the same study is cited in support of an optimal pH value of 1 for the reaction. How can a factor with no influence be optimized?

Author`s comment

The authors agree with the reviewer. The sentence *"The pH during the heating process was investigated even if pH = 1 was found as an optimal pH for the PFBHA derivatisation (Rodigast et al., 2015)."* (Line 201) is misleading and was changed to *"The pH during the heating process was investigated as well."*

10) Line 232: This is an important statement, but it leaves the reader wondering which other carbonyl compounds were being monitored in this work. It would be very helpful if the authors would list these compounds.

Author´s comment

In the present study, no other carbonyl compound was identified during the analysis with GC/MS after PFBHA derivatisation. However, according to the literature studies other carbonyl compounds can be expected as particle-phase products, e.g., propionaldehyde (Cocker et al., 2001), glyoxal (Cocker et al., 2001; Huang et al., 2015), 2-methyl-4-oxo-2-pentenal (Healy et al., 2008; Huang et al., 2014), glycolaldehyde (Cocker et al., 2001) and 3,5-dimethylbenzaldehyde (Huang et al., 2014). These compounds were not detected with the used analysis method and under the experimental conditions applied in the present study.

The sentence (Page 7, Line 230) was changed to *"According to the literature studies other carbonyl compounds can be expected as particle-phase products, e.g., propionaldehyde (Cocker et al., 2001), glyoxal (Cocker et al., 2001; Huang et al., 2015), 2-methyl-4-oxo-2-pentenal (Healy et al., 2008; Huang et al., 2014), glycolaldehyde (Cocker et al., 2001) and 3,5-dimethylbenzaldehyde (Huang et al., 2014). Noticeably, no carbonyl compounds other than methylglyoxal were identified, which showed an increase after thermal decomposition."*

11) Line 236: Confusing statement. According to Table 3, the method was developed using filters from experiments 2 and 3 from laboratory 1,3,5-TMB oxidation studies (plus methylglyoxal monomer standards). Now the method is applied to more of the samples from the same set of studies.

Author´s comment

To clarify this the sentence will be changed.

The method was developed with filter samples from experiment #2 and #3 to use laboratory-generated SOA. After the method development the quantification method was applied to filter samples from further experiments (# 1, 4, 5, 6) to investigate the influence of seed particle acidity and relative humidity on the oligomer content.

Line 234: The sentence was changed to *"[…] developed quantification method was afterwards applied to laboratory-generated SOA formed during further oxidation experiments of 1,3,5-TMB to investigate the influence of seed particle acidity and relative humidity on the oligomer content."*

12) Line 257: From the information given it is hard to see how these two studies "are in good agreement" with the present study, since the present study is measuring no acidity effect while the two literature studies compared seed / no seed conditions.

Author´s comment

The authors agree with the reviewer and delete the sentence (Page 7, Line 255) *"This is in good agreement with the studies by Cocker et al. (2001) and Wyche et al. (2009) which observed no differences of SOA formation in the presence or absence of seed particles during the photooxidation of 1,3,5-TMB with $NO_x$."*. The mentioned literature studies compared experiments with and without seed particles while in no-seed experiments SOA formation occurs due to nucleation forming organic particles. Organic particles might lead to a stronger partitioning of organic compounds and thus cannot be compared (Spittler et al., 2006; Pankow, 1994).

13) Line 289: The authors' measurements of the percentage methylglyoxal in 1,3,5-TMB SOA are right in line with other published values. Did Healy et al and Cocker et al. use seed particles? Can the authors comment on whether such percentages are consistent with partitioning theory and the vapor pressure of methylglyoxal?

Author´s comment

Healy et al., 2008 used no seed particles while Cocker et al., 2001 used $(NH_4)_2SO_4$ seed particles.

To calculate the theoretically possible methylglyoxal mass in the particle phase (according to the partitioning theory and the vapor pressure), yields of methylglyoxal in the gas phase reported in the literature have to be assumed because the gas-phase yields of methylglyoxal were not measured in the present study. In the literature it is known that the oxidation of 1,3,5-TMB results in yields of 60 – 90% of methylglyoxal in the gas phase (Smith et al., 1999; Bandow and Washida, 1985; Tuazon et al., 1986). These gas-phase yields were taken as basis for the calculation of the theoretical possible methylglyoxal present in the particle phase:

Example: $NH_4HSO_4$ seed particle; RH = 50%: Calculation of the mass of methylglyoxal present in the particle phase in case of a gas phase yield of 60%:

Consumed 1,3,5-TMB: $\Delta HC = 55$ ppb

Methylglyoxal yield of 60% $\rightarrow$ 33.0 ppb methylglyoxal $= 8.12 \times 10^{17}$ molecules m$^{-3}$

Calculation of methylglyoxal in per cent by volume with the Loschmidt constant $(2.46 \times 10^{25}$ molecules m$^{-3})$

$$\frac{8.12 \times 10^{17} \text{ molecules m}^{-3}}{2.46 \times 10^{25} \text{ molecules m}^{-3}} = 3.30 \times 10^{-8} = 3.30 \times 10^{-6} \%$$

Calculation of the partial pressure (P) of methylglyoxal:

$$\frac{100 \%}{1 \text{ atm}} = \frac{3.30 \times 10^{-6} \%}{P}$$

$$P = 3.30 \times 10^{-8} \text{atm}$$

Calculation of the concentration in the aqueous particle phase ($c_L$):

$$H = \frac{c_L}{P} \qquad H \text{ (Henry constant)} = 3.7 \times 10^3 \text{ M atm}^{-1} \text{ (Betterton and Hoffmann, 1988)}$$

$$c_L = P \times H = 3.30 \times 10^{-8} \text{ atm} \times \left(3.7 \times 10^3 \text{ M atm}^{-1}\right) = 0.00012 \text{ mol L}^{-1}$$

Related to the liquid water content (LWC) of the particles (RH = 50%; $NH_4HSO_4$ seed particles, calculated with E-AIM model III):

$$LWC = 6.66 \text{ µg m}^{-3} = 6.66 \times 10^{-9} \text{ kg in in 1 m}^3 \text{ particles}$$

Calculation of the water volume:

$$V = \frac{m}{\rho} = \frac{6.66 \times 10^{-9} \text{ kg}}{1 \text{ kg m}^{-3}} = 6.66 \times 10^{-9} \text{ m}^3 = 6.66 \times 10^{-6} \text{ L}$$

m: mass of water

ρ: density of water

Calculation of methylglyoxal in mol (with $c_L = 0.00012$ mol L$^{-1}$) in $6.66 \times 10^{-6}$ L water:

$$\frac{0.00012 \text{ mol}}{1 \text{ L}} = \frac{x}{6.66 \times 10^{-6} \text{ L}}$$

$$x = 7.99 \times 10^{-10} \text{ mol} = 7.99 \times 10^{-4} \text{ μmol}$$

Calculation of methylglyoxal in g with a molar mass of 72 g mol$^{-1}$:

$$7.99 \times 10^{-4} \text{ μmol} = \underline{\mathbf{0.06 \text{ μg}}}$$

Table 1: Calculation of the mass [μg] of methylglyoxal in the particle phase for RH = 50%.

| Theoretical gas phase methylglyoxal yield | NH$_4$HSO$_4$ | | (NH$_4$)$_2$SO$_4$/H$_2$SO$_4$ | |
| --- | --- | --- | --- | --- |
| | Calculated | Measured | Calculated | Measured |
| 60% | 0.06 μg | 0.14 μg | 0.04 μg | 0.09 μg |
| 90% | 0.09 μg | | 0.06 μg | |

Table 2: Calculation of the mass [μg] of methylglyoxal in the particle phase for RH = 75%.

| Theoretical gas phase methylglyoxal yield | NH$_4$HSO$_4$ | | (NH$_4$)$_2$SO$_4$/H$_2$SO$_4$ | |
| --- | --- | --- | --- | --- |
| | Calculated | Measured | Calculated | Measured |
| 60% | 0.11 μg | 0.08 μg | 0.11 μg | 0.09 μg |
| 90% | 0.16 μg | | 0.16 μg | |

As can be seen, at RH = 50% the calculated mass of methylglyoxal in the particle phase is lower than the measurements. In contrast, with RH = 75% the calculated mass is higher than the measured values. There are different reasons possible leading to the discrepancies. The theoretical calculations include no further reactions of methylglyoxal in the particle phase like oligomerisation, which might lead to a lower measured methylglyoxal mass. In addition, the equilibrium between the methylglyoxal monomers and oligomers can be influenced due to experimental conditions and the extraction of the filter samples with water, which is also not included in the calculations leading to lower or higher methylglyoxal masses in the particle phase.

14) Line 301: Do these literature studies suggest that either of the two explanations offered in the previous sentence are more likely than the other?

Author´s comment

The literature studies were cited because they investigate the formation of methylglyoxal oligomeric compounds and was not related to studies investigating the effect of RH on oligomer formation.

The sentence (Page 9, Line 300) was changed to *"The formation of oligomeric compounds from methylglyoxal has been investigated […]."*.

15) Line 315: The phrase "opposite trend" was unclear. What trend is this trend the opposite of?

Author´s comment

The term "opposite trend" describes the dependency of the oligomer fractions in SOA on the relative humidity with different seed particles. It can be seen in Fig. 5b, with $NH_4HSO_4$ seed particles the fraction of the oligomeric compounds in SOA decreases with increasing RH while using $(NH_4)_2SO_4/H_2SO_4$ seed particles the oligomer fraction increases under elevated RH. Thus, there is an opposite trend of the contributions of oligomeric compounds to SOA between the different seed particles under various RH values.

Line 312: The sentence was changed to *"[…] opposite trend of the oligomer fractions with RH between $NH_4HSO_4$ and $(NH_4)_2SO_4/H_2SO_4$ seed particles could be different oligomer formation mechanism caused by different seed particle acidity."*

16) Line 334: Doesn't this logic also apply to the next two reactions that involve water loss? I am unsure why aldol reactions are enhanced and esterification and imidazole formation are decreased at high RH, when all involve water loss as part of the mechanism.

Author´s comment

The water loss of aldol condensation (Figure 1) is an irreversible reaction while esterification, imidazole formation and acetal/hemiacetal formation is a reversible reaction (Figure 2).

Fig. 1: Acid-catalysed aldol-condensation (R: organic rest or H-atom).

Fig. 2: Acid-catalysed acetal/hemiacetal formation (R: organic rest or H-atom).

The following sentences were changed to:

Page 9, Line 335 *"[…] involves the reversible loss of water […]"*

Page 10, Line 336 *"[…] decreases due to the shift of the equilibrium towards the monomers."*

Technical Corrections

Line 194: "might be" should be "are". The authors know that $H^+$ and $NH_4^+$ ions are present in the samples and therefore the aqueous extracts.

Author´s comment

The sentence (Page 6, Line 195) was changed to *"[…] these ions are present […]"*

Line 205: "PFTFE" should be "PTFE"

Author´s comment

The sentence (Page 6, Line 205) was changed to *"[…] with PTFE filters,[…]"*

Line 253: "condensate" should be "condense"

Author´s comment

The sentence (Page 7, Line 251) was changed to *"[…] products condense on the pre-existing […]"*

Line 314: The period is missing after "conditions"

Author´s comment

The sentence (Page 9, Line 310) was changed to *"[…] dry conditions (RH = 0%)."*

Line 343: "is" should be "it"

Author´s comment

The sentence (Page 10, Line 340) was changed to *"[…] thus it does not provide […]"*

Line 352: "increases" should be "increase"

Author´s comment

The sentence (Page 10, Line 348) was changed to *"[…] particle phase might increase, […]"*

Table 6 does not provide any additional information beyond Figure 5b, and could be eliminated.

Author´s comment

Table 6 was deleted from the manuscript.

**The following changes were made to the manuscript**

The term *"(methylglyoxal) oligomers"* was changed to *"heat-decomposable (methylglyoxal) oligomers"* in the manuscript and the figure caption of Figure 5.

Page 2, Line 19: *"Aromatic compounds represent a large fraction of the total emitted hydrocarbon mass contributing such as up to 52% to the hydrocarbon mass in China (Liu et al., 2008)."* was changed to *"Aromatic compounds represent a large fraction of the emitted hydrocarbons contributing with up to 52% to the total non-methane hydrocarbon mass at an industrial site in china (Liu et al., 2008)."*

Page 3, Line 72: *"Thus, the present study presents a foundation approach for a reliable quantification of methylglyoxal oligomers in laboratory-generated SOA."* was changed to *"Thus, the present study presents a fundamental approach for a reliable quantification of methylglyoxal oligomers in laboratory-generated SOA."*

Page 3, Line 73: The paragraph *"The method is applicable for all oligomeric compounds, which can be decomposed into methylglyoxal monomers during the heating process at a temperature of 100 °C. As the oligomerisation mechanisms leading to the quantified oligomeric compounds are not known, it cannot be specified if the oligomers are reversibly or irreversibly formed. In addition, it cannot be excluded that there exist oligomers, which are not decomposable into their methylglyoxal monomers through the heating process. Thus, the quantified oligomers are termed as heat-decomposable methylglyoxal oligomers."* was included in the manuscript.

Page 3, Line 91: *"As OH-radical source the ozonolysis of tetramethylethylene (TME) was used (Berndt and Böge, 2006)"* was changed to *"In order to investigate OH-radical oxidation of 1,3,5-TMB at low $NO_x$ levels (< 1 ppb) and under dark conditions the ozonolysis of tetramethylethylene (TME) was used as OH-radical source (Berndt and Böge, 2006)."*

Page 3, Line 92: The paragraph *"The cycloaddition of ozone to TME yields a primary ozonide, which reacts further and forms a stabilised Criegee Intermediate (sCI). The sCI decomposes via the hydroperoxide channel, leading to the formation of OH radicals (Gutbrod et al., 1996) with a yield of 0.92 ± 0.08 (Berndt and Böge, 2006)."* was added.

Page 4, Line 100: *"To collect the particle phase after the experiments 1.2 $m^3$ of the chamber volume was sampled on a PTFE filter (borosilicate glass fiber filter coated with fluorocarbon, 47 mm in diameter, PALLFLEX T60A20, PALL, NY, USA) connected to a denuder (URG-2000-30B5, URG Corporation, Chapel Hill, NC, USA) (Kahnt et al., 2011) to avoid artefacts caused by adsorption of gas-phase organic compounds onto the filter."* was changed to *"To collect the particle phase after the experiments, 1.2 $m^3$ of the chamber volume was sampled on a PTFE filter (borosilicate glass fiber filter coated with fluorocarbon, 47 mm in diameter, PALLFLEX T60A20, PALL, NY, USA) connected to a XAD-4 coated denuder (URG-2000-30B5, URG Corporation, Chapel Hill, NC, USA; Kahnt et al., 2011) to avoid artefacts caused by adsorption of gas-phase organic compounds onto the filter."*

Page 6; Line 185: *"To probe this hypothesis a 6.25 µmol $L^{-1}$ methylglyoxal standard was heated for 15 hours and measured immediately (like sample a) or, alternatively, was allowed to stand at room temperature for 9 hours (like sample b)."* was changed to *"To probe this hypothesis a 6.25 µmol $L^{-1}$ methylglyoxal standard was heated for 15 hours and measured immediately (like sample b) or, alternatively, was allowed to stand at room temperature for 9 hours (like sample a)."*

*Page 6, Line 195: "Naturally, both of these ions might be present in the aqueous filter extract."* was changed to *"Naturally, both of these ions are present in the aqueous filter extract."*

Page 6, Line 195: The sentence *"Despite the oligomerisation of methylglyoxal monomers during the 9 hours at room temperature, the derivatisation proceeds as well during this time leading to higher methylglyoxal concentrations in sample a than in sample b, which was directly measured after 15 hours heating (Fig. 1b).*" was added.

Page 6, Line 200: The sentence "*Thus, it can be speculated that the low methylglyoxal concentrations in the filter samples are a result of further reactions with particle-phase species, which do not exist in the standard samples."* was included.

Page 6, Line 201: "*The pH during the heating process was investigated even if pH = 1 was found as an optimal pH for the PFBHA derivatisation (Rodigast et al., 2015)"* was changed to "*The pH during the heating process was investigated as well."*

Page 6, Line 205: *"The effect of the pH was examined with PFTFE filters, which were sampled after the OH-radical oxidation of 1,3,5-TMB at RH = 50% in the presence of NH₄HSO₄ particles (experiment #2)."* was changed to *"The effect of the pH was examined with PTFE filters, which were sampled after OH-radical oxidation of 1,3,5-TMB at RH = 50% in the presence of NH₄HSO₄ particles (experiment #2)."*

Page 7, Line 230: *"Noticeably, among the investigated filter samples no other carbonyl compounds showed an increase after thermal decomposition indicating that oligomers present in the particle phase of 1,3,5-TMB oxidation are solely methylglyoxal oligomers and/or oligomers of other carbonyl compounds need different conditions for decomposition."* was changed to *"According to the literature studies other carbonyl compounds can be expected as particle-phase products, e.g., propionaldehyde (Cocker et al., 2001), glyoxal (Cocker et al., 2001; Huang et al., 2015), 2-methyl-4-oxo-2-pentenal (Healy et al., 2008; Huang et al., 2014), glycolaldehyde (Cocker et al., 2001) and 3,5-dimethylbenzaldehyde (Huang et al., 2014). Noticeably, no carbonyl compounds other than methylglyoxal were identified, which showed an increase after thermal decomposition."*

Page 7, Line 234: *"The developed method was afterwards applied to laboratory-generated SOA formed by the oxidation of 1,3,5-TMB."* was changed to "The *developed quantification method was afterwards applied to laboratory-generated SOA formed during further oxidation experiments of 1,3,5-TMB to investigate the influence of seed particle acidity and relative humidity on the oligomer content."*

Page 7, Line 251: *"These products condensate on the pre-existing seed particles resulting in the immediate particle growth observed in Fig. 4a."* was changed to *"These products condense on the pre-existing seed particles resulting in the immediate particle growth observed in Fig. 4a."*

Page 7, Line 255: The sentence *"This is in good agreement with the studies by Cocker et al., 2001 and Wyche et al., 2009 which observed no differences of SOA formation in the presence or absence of seed particles during the photooxidation of 1,3,5-TMB with NOₓ."* was deleted.

Page 8, Line 259: The paragraph *"The RH value can have an influence on the phase state of the particles while the phase state has an effect on the partitioning of the compounds into the particles and the particle-phase reactions (Ziemann, 2010; Saukko et al., 2012). In the present study the particles are a mixture of inorganic and partitioned organic compounds, thus the phase state is not known but it might be possible that the phase state influences ΔM and the SOA yields."* was added to the manuscript.

Page 9, Line 300: *"This has been investigated in a number of studies (e. g. De Haan et al., 2009; Kalberer et al., 2004; Loeffler et al., 2006; Zhao et al., 2006; Sareen et al., 2010; Altieri et al., 2008)."* was changed to *"The formation of oligomeric compounds from methylglyoxal has been investigated in a number of studies (e. g. De Haan et al., 2009; Kalberer et al., 2004; Loeffler et al., 2006; Zhao et al., 2006; Sareen et al., 2010; Altieri et al., 2008)."*

Page 9, Line 306: *"The concentrations were converted into the fraction of methylglyoxal oligomers of $\Delta M$ using the molar mass of methylglyoxal (Mw = 72.06 g mol$^{-1}$; Table 6)."* was changed to *"The concentrations were converted into the fraction of methylglyoxal oligomers of $\Delta M$ using the molar mass of methylglyoxal (Mw = 72.06 g mol$^{-1}$)."*

Page 9, Line 310: *"In the presence of $NH_4HSO_4$ seed particles the highest oligomer fraction (8.2 ± 0.7%) can be observed with RH = 0% whereas in the presence of $(NH_4)_2SO_4/H_2SO_4$ seed particles the oligomer fraction is the lowest (2.1 ± 0.4%) under dry conditions."* was changed to *"In the presence of $NH_4HSO_4$ seed particles the highest oligomer fraction (8.2 ± 0.7%) can be observed with RH = 0% whereas in the presence of $(NH_4)_2SO_4/H_2SO_4$ seed particles the oligomer fraction is the lowest (2.1 ± 0.4%) under dry conditions (RH = 0%)."*

Page 9, Line 312: *"A possible explanation for the opposite trend could be a different oligomer formation mechanism dependent on the different seed particles."* was changed to *"A possible explanation for the opposite trend of the oligomer fractions with RH between $NH_4HSO_4$ and $(NH_4)_2SO_4/H_2SO_4$ seed particles could be different oligomer formation mechanism caused by different seed particle acidity."*

Page 9, Line 335: *"This equilibrium reaction involves the loss of water as it was reported for acetal/hemiacetal formation (Lim et al., 2010)."* was changed to *"This equilibrium reaction involves the reversible loss of water as it was reported for acetal/hemiacetal formation (Lim et al., 2010)."*

Page 10, Line 336: *"Thus, it can be expected that with higher LWCs the contribution of esterification reactions to oligomer formation decreases."* was changed to *"Thus, it can be expected that with higher LWCs the contribution of esterification reactions to oligomer formation decreases due to the shift of the equilibrium towards the monomers."*

Page 10, Line 340: *"However, imidazole formation involves also a loss of water, thus is does not provide a feasible explanation for the higher oligomer fraction at higher RH with $(NH_4)_2SO_4/H_2SO_4$ seed particles."* was changed to *"However, imidazole formation involves also a reversible loss of water, thus it does not provide a feasible explanation for the higher oligomer fraction at higher RH with $(NH_4)_2SO_4/H_2SO_4$ seed particles."*

Page 10, Line 348: *"Nevertheless, it can be expected, that with higher LWC of the seed particles and thus with a higher reaction volume, the absolute amount of methylglyoxal in the particle phase might increases, but not its particle-phase concentrations."* was changed to *"Nevertheless, it can be expected, that with higher LWCs of the seed particles and thus with a higher reaction volume, the absolute amount of methylglyoxal in the particle phase might increase, but not its particle-phase concentration."*

Page 12, Line 467: The reference Gutbrod et al., 1996 *"Gutbrod, R., Schindler, R. N., Kraka, E., and Cremer, D.: Formation of OH radicals in the gas phase ozonolysis of alkenes: the unexpected role of carbonyl oxides, Chemical Physics Letters, 252, 221-229, 10.1016/0009-2614(96)00126-1, 1996."* was included in the manuscript.

Page 13, Line 502: The reference Huang et al., 2014 *"Huang, M. Q., Hu, C. J., Guo, X. Y., Gu, X. J., Zhao, W. X., Wang, Z. Y., Fang, L., and Zhang, W. J.: Chemical composition of gas and particle-phase products of OH-initiated oxidation of 1,3,5-trimethylbenzene, Atmospheric Pollution Research, 5, 73-78, 10.5094/APR.2014.009, 2014."* was included in the manuscript

Page 15, Line 616: The reference Sauko et al., 2012 *"Saukko, E., Lambe, A. T., Massoli, P., Koop, T., Wright, J. P., Croasdale, D. R., Pedernera, D. A., Onasch, T. B., Laaksonen, A., Davidovits, P., Worsnop, D. R., and Virtanen, A.: Humidity-dependent phase state of SOA particles from biogenic and anthropogenic precursors, Atmospheric Chemistry and Physics, 12, 7517-7529, 10.5194/acp-12-7517-2012, 2012."* was included in the manuscript.

Page 16, Line 692: The reference Ziemann, 2010 *"Ziemann, P. J.: Atmospheric chemistry Phase matters for aerosols, Nature, 467, 797-798, 2010."* was included in the manuscript.

Page 19, Line 731: Table 6 was deleted.

---

## Author Comment (AC2) · 5 Jan 2017

Responses to Reviewers' Comments:

We thank the reviewers for their constructive comments. The manuscript is revised based on the suggestions made and detailed responses to the reviewers are addressed as follows.

Referee #2

Overall Comment and Recommendation:

This study is primarily a method development study aimed at quantifying oligomers derived from multiphase chemistry of methylglyoxal by using GC/MS with prior derivatization. Many groups have shown that when you heat certain SOA types, such as IEPOX-derived SOA, you measure monomeric products (like 2-methyltetrols) in high quantities (Lopez-Hilfiker et al., 2016, ES&T). This is important since monomeric products like 2-methyltetrols are too volatile to exist in such large quantities to explain the observed SOA formation in lab or field studies. The present study utilized GC/MS with PFBHA derivatization to detect methylglyoxal monomers found in 1,3,5-TMB derived SOA. The authors systematically examined the influence of heating time, pH, and heating temperature on the decomposition of methylglyoxal oligomers. The authors found that the best result was likely achieved when heating the extracts for 24 hours at 100 degrees C and at pH of 1. The authors found that the oligomers accounted for up to 8% of the total SOA mass. The method the authors develop could be very useful in trying to provide mass closure of the oligomer fraction of methylglyoxal-derived oligomers. More importantly, this method could likely be adapted to determine oligomer mass fractions in other types of SOA. This is important since we currently lack appropriate authentic standards to quantify the oligomer content of the SOA. More often we have standards available to quantify the monomers, so this method would be of interest to many research groups working in this area. Before I can recommend final publication in ACP, I have many specific comments that need to be addressed by the authors. These specific comments are outlined below. Due to the nature of these comments, I must recommend to the Editor that this manuscript be accepted with major revisions noted. There are several method details missing that need to be clarified or added to the main text. In addition, in several sections of the manuscript, the English writing is at times quite poor. As a result, I encourage the authors to conduct further editing on the writing before resubmission.

Author`s comment

The manuscript was edited again.

Specific Comments:

1) Citations in main text:

The authors cite references through the manuscript using "Last name of first author et al., Year." As an example, please refer to Page 2, Line 52. The authors should change to Kalberer et al. (2004). The style in ACP is always "Author last name et al. (year)."

Author`s comment

The authors changed the style of the references cited in the manuscript.

2) Filter Extractions:

Have the authors tested extracting the filters in an organic solvent such as acetonitrile or methanol? I wonder how the extraction efficiencies of potentially large oligomers change with extraction solvent? This factor should at least be discussed in this manuscript and the likely uncertainties in obtaining exact oligomer concentrations from this method. What I'm getting at is the authors assume in the text that the filter extraction efficiency is likely 100% in water.

Author`s comment

The authors didn´t investigate the extraction of the oligomers from the PTFE filter samples with organic solvents like acetonitrile or methanol due to the following reasons:

i)     Methylglyoxal was derivatised with PFBHA after extraction from the PTFE filters leading to the formation of oximes. The formed oximes were extracted with dichloromethane before injection in GC/MS, which was found as the best extracting reagent during optimisation of the PFBHA derivatisation method (Rodigast et al., 2015). Methanol and acetonitrile are miscible with dichloromethane, thus an extraction of the oximes would not be possible.

ii)    The boiling points of methanol and acetonitrile are lower than for water. Thus, the samples could not be heated to a temperature of 100 °C which was found to be necessary for the decomposition of oligomers.

Due to the listed reasons it might be necessary to remove the organic solvents after extraction of the oligomers from the PTFE filters. To avoid a drying step, which might lead to a loss of compounds, water was used as extracting reagent for methylglyoxal and their oligomers.

Nevertheless, because the extraction efficiency was not investigated in the present study the following paragraph is included in the manuscript (Page 4, Line 115) *"Noticeably, the extraction efficiency was not investigated in the present study, thus it is not known. Water was used as extracting reagent, because organic solvents like methanol and acetonitrile have lower*

*boiling points than water, thus lower heating temperatures can be applied for the decomposition of the oligomers. Besides this, organic solvents like methanol and acetonitrile are miscible with dichloromethane, thus an extraction of the derivatised methylglyoxal with dichloromethane prior the injection into GC/MS would not be possible."*

3) GC/MS operating details:

You should state here explicitly how long your GC/MS run is. Since it appears this is a long GC/MS run, did the authors check to see how the methylglyoxal standard calibration changes throughout the run? For example, did the authors consider rerunning the calibration at the end of the run? Did the response factor change/drift dramatically?

Author`s comment

The GC/MS method has a run time of 36 minutes including 10 minutes post run at 230 °C to remove remaining compound from the column. To mention this in the manuscript the sentence (Page 5, Line 146) was changed to *"The temperature of 230°C was held constant for 1 minute and ended with 320°C for 10 minutes, thus the method has a run time of 36 minutes."*

The 5-point calibration was measured three times before starting analysis of the samples from the aerosol chamber experiments. In addition, after six measurements of the chamber samples one calibration sample was measured again (Figure 3).

[Figure]

Figure 3: Comparison of the calibration at the beginning of the GC/MS measurements with a repeated measurement of the calibration after measuring samples from chamber experiment.

As it can be seen in Figure 3, the comparison between the calibration at the beginning of the measurements and after measuring filter samples showed good agreements. Thus, it can be concluded that the methylglyoxal calibration and their response factor had no significant changes during the analysis.

4) Method Development:

In the method development section when varying heating time, pH, and heating temperature, it seems that for the latter two the same experiment was used (i.e., experiment 2). In contrast, the heating time was explored with experiment 3. I think the authors would have been better to use the same type of aerosol generated under the exact same conditions when exploring these parameters of the method. I'm curious to know why this wasn't done or why it is justified to do this as is?

Author`s comment

To clarify this a sentence was included in the manuscript.

The heating time was investigated with filter samples from experiment #3 and the pH value as well as the heating temperature was investigated with samples from experiment #2.

For each measurement one filter half is needed. Due to the volume of the aerosol chamber (19 m³) it was not possible to collect more than four filter samples (1.2 m³ each) per experiment, which was not enough to investigate all method parameters.

For that reason, it was at least considered to use filter samples from the same experiment for the optimisation of the respective method parameters. Thus, the measurements within each method parameter are comparable and can be used for method optimisation.

The sentence (Page 4, Line 109) *"Filter samples from the same experiments were used for the optimisation of the respective method parameters."* was added.

5) SOA Yields:

Since the authors spend time in this study reporting SOA yields from 1,3,5-TMB oxidation, I have some questions about the differences in the amount of organic aerosol produced under the different RH conditions. Considering that you observe more SOA under dry conditions when compared to more humid conditions, I wonder what role your chamber walls are playing? Recent work from the Caltech (Seinfeld), CU-Boulder (Jimenez and Ziemann), and CMU (Donahue) groups suggest that the wall effect could be really important, especially if your goal is to report SOA yields in the literature. Thus, maybe you are losing more things to your wall under higher RH conditions? Have the authors considered how to correct SOA yields for this effect? If not, you should at least acknowledge the likely importance of wall losses of semivolatile and less volatile organic vapors.

Author`s comment

The wall loss can has an effect on the SOA yields as well as on the product distribution in the particle phase due to partitioning of the precursor compounds or products to the wall instead

of into the particles (Matsunaga and Ziemann, 2010; Loza et al., 2010; Zhang et al., 2014; Grosjean, 1985; McMurry and Grosjean, 1985).

In the present study, the wall loss was not investigated but due to the higher surface of the aerosol chamber LEAK in comparison to the surface of the particles the wall loss might play an important role. It was calculated that LEAK has an about 3500 fold higher surface than the particles, thus the compounds are prone for deposition to the walls.

In previous experiments the wall loss of 1,3,5-TMB was measured at RH ≈ 0% resulting in an uptake coefficient $\gamma = 8.8 \times 10^{-8}$. Such uptake coefficients have to be measured for each oxidation product of 1,3,5-TMB in dependency on relative humidity to determine their wall loss and to estimate the effect of the wall loss on the SOA yield.

Even if the wall loss of methylglyoxal was not measured it is possible that the methylglyoxal concentration in the particle phase decreases due to partitioning of gas-phase methylglyoxal to the chamber walls. The lower particle-phase concentration of methylglyoxal results in lower fractions of methylglyoxal oligomers in SOA and thus to lower SOA yields. In addition, it could be assumed that the wall loss increases with increasing RH as it was measured for glyoxal by Loza et al., 2010.

Nevertheless, the wall loss was not considered during the investigation of the SOA yields of 1,3,5-TMB. Thus, it might be possible that the enhanced wall loss is a reason for the lower SOA yields under high RH values. To mention this in the manuscript the following sentence was included (Page 7, Line 246): *"Noticeably, the SOA yields of 1,3,5-TMB are not corrected for the wall loss to the surface of the aerosol chamber, which might have an influence on the reported SOA yields."*

6) Page 9, Lines 298-299:

Could it be that methylglyoxal's chamber wall losses are also changing with RH? Did the authors consider injecting methylgloxal in the gas phase of the chamber and investigate its wall losses with different RHs? That might provide more insights into the importance of your chamber wall.

Author`s comment

It might be possible that methylglyoxal has an increasing wall loss with increasing RH values. This was not investigated in the present study thus, the importance of the wall loss of methylglyoxal during the oxidation of 1,3,5-TMB cannot be estimated. The investigation of this effect would be an additional project and was not the focus of the study.

7) Page 9, Lines 329-332:

I wonder how the different seed aerosols you use might cause differences in aerosol phase separation/morphology? What role could this potentially have in explaining the differences in the oligomer fraction?

Author`s comment

$NH_4HSO_4$ and $(NH_4)_2SO_4/H_2SO_4$ seed particles might have different phase states and thus different phase separations and morphologies under various RH values. The phase state is influenced due to deliquescence and efflorescence of the seed particles. The deliquescence and efflorescence RH of the pure inorganic particles can be found in the literature. The deliquescence RH of $(NH_4)_2SO_4$ particles was reported to be 79% while for $NH_4HSO_4$ particles a value of 39% was found (Cziczo et al., 1997). Thus, pure $NH_4HSO_4$ seed particles are liquid under the investigated RH range of 50% and 75% while $(NH_4)_2SO_4$ seed particles are solid.

As it was answered to comment 4 from reviewer #1, organic compounds in the particles influence the deliquescence point (Andrews and Larson, 1993; Lightstone et al., 2000) as well as the hygroscopicity of the particles (Lightstone et al., 2000; Prenni et al., 2003; Chen and Lee, 1999). In the present study mixed particles including inorganic and organic compounds exist, thus the phase separation/morphology of the particles in the oxidation experiments of 1,3,5-TMB may differ from the pure inorganic particles indicating the phase state of the particles during the oxidation experiments is not known.

In general, it is estimated that the phase of the particles can have an influence on the partitioning of the compounds from the gas into the particle phase or on further reactions in the particle phase (Ziemann, 2010; Saukko et al., 2012). In case of oligomerisation the oligomer fraction in the particles can be influenced due to the partitioning of methylglyoxal monomers into the particles or the direct formation of methylglyoxal in the particles as well as the further reaction of particulate methylglyoxal forming oligomers.

Virtanen et al., 2010 found an amorphous solid phase state of particles including oligomers. It was assumed, that the further reactions in the particle phase might be inhibited in solid particles (Saukko et al., 2012) thus, further oligomerisation reactions can be lowered after a certain fraction of oligomers exist in the particles. In addition, the partitioning of methylglyoxal monomers can be inhibited into solid particles (Saukko et al., 2012), which might also lead to lower oligomer fractions in SOA. It is not known if methylglyoxal is formed in the gas phase and partition into the particles or if it is formed directly in the particle phase. Under the assumption that particle-phase reactions are inhibited in solid particles it might also be possible

that the formation of methylglyoxal in the particle phase is lowered in solid particles leading to lower oligomer fractions as well.

As the phase state of the particles in the present study is not known, the influence of the phase separation/morphology on the oligomerisation reactions is very speculative but it can be assumed that it has an influence.

The following paragraph was included in the manuscript (Page 8, Line 259): *"The RH value can have an influence on the phase state of the particles while the phase state has an effect on the partitioning of the compounds into the particles and the particle-phase reactions (Ziemann, 2010; Saukko et al., 2012). In the present study the particles are a mixture of inorganic and partitioned organic compounds, thus the phase state is not known but it might be possible that the phase state influences ΔM and the SOA yields."*

8) Oligomer Types:

It appears that the authors only consider the oligomer type resulting from methylglyoxal + methylglyoxal type reactions. However, considering the plethora of other monomers when oxidizing a VOC like 1,3,5-TMB, why did the authors not consider other types of oligomer reactions involving methylgloyxal + some other oxidized product? Was there no evidence for this in your GC/MS data? Related to this, why didn't the authors provide a TIC or EIC in the main text? In either the TIC or EIC, it would be helpful to provide peak labels and likely respective mass spectra to each chromatographic peak.

I mention this as recent work by Lin et al. (2014, ES&T) demonstrated the varying types of IEPOX-derived oligomers under different RH and seed aerosol conditions. It appeared from their LC/MS data that there was a very wide degree of types (e.g., light versus non-light absorbing) and lengths of oligomers present.

Author`s comment

The authors agree with the reviewer that other compounds than methylglyoxal can contribute to oligomer formation. But, in the present study only those oligomers are considered, which can be decomposed into methylglyoxal during the heating process. Notably, the applied GC/MS method is suitable for a variety of carbonyl compounds (Rodigast et al., 2015; Figure 4).

[Figure]

Figure 4: TIC of an unheated and heated filter extract; a: cyclohexanone-2,2,6,6,-d4, b: 2-trifluoronethyl)benzaldehyde, c: methylglyoxal.

The peaks labeled with "a" and "b" are the internal standards cyclohexanone-2,2,6,6,-d4 and 2-trifluoromethylbenzaldehyde. Methylglyoxal is labeled with "c" and it can be seen, that the peak intensity increases after heating due to decomposition of the methylglyoxal oligomers into methylglyoxal.

There are additional peaks, e.g., at a retention time of 16.8 minutes with a mass to charge ratio of 333 [M˙⁺]. It was not possible to identify this compound during the GC/MS analysis.

To mention this in the manuscript the sentence (Page 7, Line 230) was changed to *"[…] no carbonyl compounds other than methylglyoxal were identified, which showed an increase after thermal decomposition."*

Minor Comments:

1) Introduction, Lines 36-42: When generalizing the oligomeric mechanisms leading to SOA, why not include those derived from acid-catalyzed hydrolysis of epoxides (e.g.,Paulot et al., 2009, Science, Surratt et al., 2010, PNAS; Lin et al., 2014, ES&T)?

Author`s comment

The acid-catalysed hydrolysis of epoxides will also be included in the manuscript. Thus the sentence (Page 2, Line 36) was changed to: *"[…] imine formation (Altieri et al., 2008; Sato et al., 2012; Tan et al., 2010; De Haan et al., 2011; Sedehi et al., 2013), hydrolysis of epoxides (Paulot et al., 2009; Surratt et al., 2010) […]"*

2) Page 2, Line 46: You should probably define the ESI/MS/MS acronym being associated with tandem mass spectrometry interfaced to ESI.

Author`s comment

The sentence (Page 2, Line 43) was changed to: *"[…] electrospray ionisation mass spectrometry and electrospray ionisation tandem mass spectrometry (ESI/MS, ESI/MS/MS).".*

3) Page 2, Line 50: Probably change "the effort for structure" to "past efforts for structural"

Author`s comment

The sentence (Page 2, Line 50) was changed to *"[…] the past effort for structure elucidation […]".*

4) Page 3, Line 72: Do you mean to say "fundamental" here instead of "foundation"?

Author`s comment

The sentence (Page 3, Line 72) was changed to *"[…] presents a fundamental approach […]".*

5) Page 3, Line 87: should this say instead: "was investigated in the Leipziger AerosolKammer (LEAK) chamber"?

Author`s comment

The sentence (Page 3, Line 87) was changed to *"[…] in the LEipziger AerosolKammer (LEAK).".*

6.) Page 3, Line 90: Maybe change "ammonium hydrogensulfate" to "ammonium bisulfate?

Author`s comment

The sentence (Page 3, Line 89) was changed to *"[…] of ammonium bisulfate […]".*

7) Page 3, Line 90: Maybe change "ammoniumsulfate" to "ammonium sulfate"?

Author`s comment

The sentence (Page 3, Line 89) was changed to *"[…] or ammonium sulfate […]".*

8) Page 4, Line 98: Add vendor and model to "(PTR-TOF MS)"

Author`s comment

The sentence (Page 4, Line 96) was changed to *"[…] flight mass spectrometer (PTR-TOF MS; 8000; IONICON Analytik, Innsbruck, Germany) […]".*

9) Page 4, Line 99: Add vendor and model to "(SMPS)"

Author`s comment

The sentence (Page 4, Line 98) was changed to *"[…] scanning mobility particle sizer (SMPS; 3010, TSI, USA) […]"*.

10) Page 4, Line 99: Why do the authors use 1 g cm$^{-3}$ density? Are you using this based on a previous study? If so, please justify why you used this aerosol density.

Author´s comment

As it was mentioned in the answer to comment 5 from reviewer #1, the density of 1 g cm$^{-3}$ is not based on previous measurements. Different values can be found in the literature while mostly a density of 1.4 g cm$^{-3}$ (Sato et al., 2012; Praplan et al., 2014; Müller et al., 2012) based on the measurements by Alfarra et al., 2006 was used. In contrast, Kleindienst et al., 1999 used a density of 1 g cm$^{-3}$, which shows the discrepancies between the literature studies. In the present study a density of 1 g cm$^{-3}$ was used, which was mentioned in the publication, thus a recalculation with other density values is possible.

11) Page 4, Line 100: Change "the particle phase" to "aerosol"

Author´s comment

The term "particle phase" was not changed to "aerosol" because only the particle phase is sampled on the PTFE filters and thus the term "aerosol" can be misleading.

12) Page 4, Line 100: Insert comma between "experiments" and "1.2"

Author`s comment

The sentence (Page 4, Line 100) was changed to *"[…] after the experiments, 1.2 m³ of the chamber volume […]"*.

13) Page 4, Line 104: Did the authors determine what the break through could be on these filters during experiments? Were control tests done to know how well the denuder worked?

Author`s comment

In the present study the break through of the particle-phase products on the denuders was not investigated. This have to be investigated for each single product as it was done by Kahnt et al., 2011.

14) Page 7, Line 252: Change "condensate" to "condense"

Author`s comment

The sentence (Page 7, Line 251) was changed to *"[…] products condense on the […]"*.

15) Page 8, Lines 279-281: Citation is needed here. Are the authors arguing that particle-phase acidity might also be required for methylglyoxal oligomers to form? If so, is this why you think LWC matters? That is, the higher the LWC the more likely the aerosol pH is less acidic and thus affecting the amount of SOA due to oligomer formation? This is unclear to me in the current text.

Author`s comment

The structure of the oligomers is not clear, thus it is unknown, which reactions contribute to oligomer formation and if these are acid-catalysed or not. In the literature studies by Yasmeen et al., 2010; Sedehi et al., 2013 and Sareen et al., 2010 it is assumed that the pH value has an effect on the oligomerisation reactions. These findings were also used in the present study to explain the dependency of the oligomer fraction in SOA from the type of seed particle. Thus, due to the present study it was possible to support the findings by Yasmeen et al., 2010; Sedehi et al., 2013 and Sareen et al., 2010.

The LWC has an influence on the pH value as well as on the equilibrium of the oligomerisation reactions. Thus, it is not clear if the LWC effects the oligomer fraction of SOA due to changing the pH of the seed particles or the equilibrium state of the oligomerisation reaction.

To further verify this the structure of the oligomers and thus, the formation mechanism have to be elucidated, which was not the focus of the present study.

The sentence (Page 8, Line 279) is changed to *"Additionally, it can be speculated that the formation […]"*.

16) Page 9, Line 308: Do the authors mean to say "on average ~2%"?

Author`s comment

The sentence (Page 9, Line 308) was changed to: *"An average oligomer fraction of ≈ 2% up to ≈ 8% was observed."*.

17) Page 10, Line 350: Change "increases" to "increase"

Author`s comment

The sentence (Page 10, Line 348) was changed to *"[…] particle phase might increase, but […]"*.

The following changes were made to the manuscript

The style of the references are changed to *"Author last name et al. (year)"* in the manuscript.

Page 1, Line 2: *"Methylglyoxal is often described to form oligomeric compounds in the aqueous particle phase which might have a significant contribution to the formation of aqueous secondary organic aerosol (aqSOA)."* was changed to *"Methylglyoxal forms oligomeric compounds in the atmospheric aqueous particle phase, which could establish a significant contribution to the formation of aqueous secondary organic aerosol (aqSOA)."*

Page 1, Line 9: *"The method development was focused on the heating time (varied between 15 and 48 hours), pH during the heating process (pH = 1 - 7), and heating temperature (50°C, 100°C) and optimised values for these conditional parameters are presented."* was changed to *"The method development was focused on the heating time (varied between 15 and 48 hours), pH during the heating process (pH = 1 - 7), and heating temperature (50°C, 100°C). The optimised values of these method parameters are presented."*

Page 1, Line 12: *"The developed method was applied to quantify methylglyoxal oligomers formed during the OH-radical oxidation of 1,3,5-TMB in the Leipziger aerosol chamber (LEAK)."* was changed to *"The developed method was applied to quantify heat-decomposable methylglyoxal oligomers formed during the OH-radical oxidation of 1,3,5-trimethylbenzene (TMB) in the Leipziger aerosol chamber (LEAK)."*

Page 1, Line 14: *"A fraction of methylglyoxal oligomers of up to 8% of the produced organic particle mass was found, highlighting the importance of those oligomers formed solely by methylglyoxal for SOA formation."* was changed to *"A fraction of heat-decomposable methylglyoxal oligomers of up to 8% in the produced organic particle mass was found, highlighting the importance of those oligomers formed solely by methylglyoxal for SOA formation."*

Page 2, Line 20: *"One of these aromatic compounds is 1,3,5-trimethylbenzene (1,3,5-TMB), which was measured in the gas phase in concentrations ranging from 0.7 to 40.6 µg m$^{-3}$ (Gee and Sollars, 1998; Khoder, 2007)."* was changed to *"One of these aromatic compounds is 1,3,5-trimethylbenzene (TMB), which was measured in the gas phase in concentrations ranging from 0.7 to 40.6 µg m$^{-3}$ (Gee and Sollars, 1998; Khoder, 2007)."*

Page 2, Line 22: *"1,3,5-TMB can be oxidised in the gas phase leading to low-volatile oxidation products which partition into the particle phase and form secondary organic aerosol (SOA)."*

was changed to *"The gas-phase oxidation of 1,3,5-TMB leads to low-volatile oxidation products, which partition into the particle phase and form secondary organic aerosol (SOA)."*

Page 2, Line 24: *"Oxidation products of 1,3,5-TMB were investigated in a number of literature studies (e. g. Huang et al., 2015; Baltensperger et al., 2005; Kalberer et al., 2004; Kalberer et al., 2006; Paulsen et al., 2005; Healy et al., 2008; Cocker et al., 2001; Smith et al., 1999; Metzger et al., 2008; Wyche et al., 2009; Yu et al., 1997) and methylglyoxal was found as an oxidation product (Metzger et al., 2008; Healy et al., 2008; Cocker et al., 2001; Smith et al., 1999; Wyche et al., 2009; Baltensperger et al., 2005; Rickard et al., 2010; Kalberer et al., 2004; Yu et al., 1997; Kleindienst et al., 1999; Müller et al., 2012; Nishino et al., 2010; Hamilton et al., 2003; Tuazon et al., 1986; Bandow and Washida, 1985) with a fraction of the particle mass of up to 2% (Healy et al., 2008; Cocker et al., 2001)."* was changed to *"Oxidation products of 1,3,5-TMB were investigated in a number of literature studies (e. g. Huang et al., 2015; Baltensperger et al., 2005; Kalberer et al., 2004; Kalberer et al., 2006; Paulsen et al., 2005; Healy et al., 2008; Cocker et al., 2001; Smith et al., 1999; Metzger et al., 2008; Wyche et al., 2009; Yu et al., 1997). Methylglyoxal was found as one of the most important oxidation product (Metzger et al., 2008; Healy et al., 2008; Cocker et al., 2001; Smith et al., 1999; Wyche et al., 2009; Baltensperger et al., 2005; Rickard et al., 2010; Kalberer et al., 2004; Yu et al., 1997; Kleindienst et al., 1999; Müller et al., 2012; Nishino et al., 2010; Hamilton et al., 2003; Tuazon et al., 1986; Bandow and Washida, 1985; Lim and Turpin, 2015) contributing with a fraction of up to 2% to the particle mass (Healy et al., 2008; Cocker et al., 2001.)"*

Page 2, Line 31: *"Methylglyoxal has often been described to form oligomeric compounds in the aqueous particle phase (see, e. g. Herrmann et al., 2015 for an overview; De Haan et al., 2009; Kalberer et al., 2004; Loeffler et al., 2006; Zhao et al., 2006; Sareen et al., 2010; Altieri et al., 2008). These oligomers are supposed to play an important role in the formation of aqueous secondary organic aerosols (aqSOA; e. g. Kalberer et al., 2004)."* was changed to *"Methylglyoxal has often been described to form oligomeric compounds in the aqueous particle phase (see, e. g. Herrmann et al., 2015 for an overview; De Haan et al., 2009; Kalberer et al., 2004; Loeffler et al., 2006; Zhao et al., 2006; Sareen et al., 2010; Altieri et al., 2008), which are supposed to play an important role in the formation of aqueous secondary organic aerosols (aqSOA; e. g. Kalberer et al., 2004)."*

Page 2, Line 31: The reference Lim and Turpin, 2015 *"Lim, Y. B., and Turpin, B. J.: Laboratory evidence of organic peroxide and peroxyhemiacetal formation in the aqueous phase and*

*implications for aqueous OH, Atmospheric Chemistry and Physics, 15, 12867-12877, 10.5194/acp-15-12867-2015, 2015."* was included.

Page 2, Line 36: *"In general, oligomeric compounds can be formed in the aqueous particle phase through aldol condensation (e. g. Tilgner and Herrmann, 2010; Sareen et al., 2010; Sedehi et al., 2013; Krizner et al., 2009; Barsanti and Pankow, 2005; De Haan et al., 2009; Yasmeen et al., 2010), acetal/hemiacetal formation (Kalberer et al., 2004 and Yasmeen et al., 2010), esterification (Altieri et al., 2008; Sato et al., 2012; Tan et al., 2010; De Haan et al., 2011; Sedehi et al., 2013), imine formation (Altieri et al., 2008; Sato et al., 2012; Tan et al., 2010; De Haan et al., 2011; Sedehi et al., 2013), polymerisation, and radical – radical reactions (Schaefer et al., 2015; Tan et al., 2012; Lim et al., 2013)."* was changed to *"In general, oligomeric compounds can be formed in the aqueous particle phase through aldol condensation (e. g. Tilgner and Herrmann, 2010; Sareen et al., 2010; Sedehi et al., 2013; Krizner et al., 2009; Barsanti and Pankow, 2005; De Haan et al., 2009; Yasmeen et al., 2010), acetal/hemiacetal formation (Kalberer et al., 2004 and Yasmeen et al., 2010), esterification (Altieri et al., 2008; Sato et al., 2012; Tan et al., 2010; De Haan et al., 2011; Sedehi et al., 2013), imine formation (Altieri et al., 2008; Sato et al., 2012; Tan et al., 2010; De Haan et al., 2011; Sedehi et al., 2013), hydrolysis of epoxides (Paulot et al., 2009; Surratt et al., 2010), polymerisation, and radical – radical reactions (Schaefer et al., 2015; Tan et al., 2012; Lim et al., 2013)."*

Page 2, Line 43: *"As it can be seen from Table 1, a number of mass spectrometric methods were used including (matrix assisted) laser desorption/ionisation mass spectrometry (MALDI-MS, LDI-MS), Fourier transform ion cyclotron resonance mass spectrometry (FT-ICR-MS), and electrospray ionisation mass spectrometry (ESI/MS, ESI/MS/MS)."* was changed to *"As it can be seen in Table 1, a number of mass spectrometric methods were used including (matrix assisted) laser desorption/ionisation mass spectrometry (MALDI-MS, LDI-MS), Fourier transform ion cyclotron resonance mass spectrometry (FT-ICR-MS), electrospray ionisation mass spectrometry and electrospray ionisation tandem mass spectrometry (ESI/MS, ESI/MS/MS)."*

Page 2, Line 50: *"Despite the effort for structure elucidation of oligomeric compounds a suitable quantification method is not available."* was changed to *"Despite the past effort for structure elucidation of oligomeric compounds a suitable quantification method is not available."*

Page 2, Line 51: *"Mostly, an overall contribution of oligomers to the particle mass was determined using e.g. a volatility tandem differential mobility analyser (VTDMA)."* was changed to *"Mostly, an overall contribution of oligomers to the particle mass was determined using, e.g., a volatility tandem differential mobility analyser (VTDMA)."*

Page 2, Line 57: *"Dommen et al., 2006 detected a contribution of oligomers increasing from 27% to 44% in the first 5 hours to organic particle mass formed in the photooxidation of isoprene."* was changed to *"Dommen et al. (2006) detected a contribution of oligomers to the organic particle mass increasing from 27% to 44% in the first 5 hours of the photooxidation of isoprene."*

Page 3, Line 62: *"De Haan et al., 2009 estimated the oligomer fraction formed by methylglyoxal in the aqueous phase concluding that 37% of methylglyoxal are dimers and oligomers with NMR."* was changed to *"De Haan et al. (2009) estimated the oligomer fraction formed by methylglyoxal in the aqueous phase with NMR concluding 37% of methylglyoxal are dimers and oligomers."*

Page 3, Line 70: *"In summary, a variety of methods exists which quantify the fraction of oligomeric compounds derived from methylglyoxal, but the results are contradicting due to the lack of a suitable method of quantification and second, due to different reaction conditions used in the studies."* was changed to *"In summary, a variety of methods exists which quantify the fraction of oligomeric compounds derived from methylglyoxal, but the results are contradicting due to the lack of a suitable method for quantification and second, due to different reaction conditions used in the studies."*

Page 3, Line 72: *"Thus, the present study presents a foundation approach for a reliable quantification of methylglyoxal oligomers in laboratory-generated SOA."* was changed to *"Thus, the present study presents a fundamental approach for a reliable quantification of methylglyoxal oligomers in laboratory-generated SOA."*

Page 3, Line 87: *"The OH-radical oxidation of 1,3,5-TMB was investigated in the aerosol chamber LEAK (Leipziger Aerosolkammer)."* was changed to *"The OH-radical oxidation of 1,3,5-TMB was investigated in the LEipziger AerosolKammer (LEAK)."*

Page 3, Line 89: *"The experiments were conducted in the presence of ammonium hydrogensulfate particles or ammoniumsulfate particles mixed with sulfuric acid to achieve different seed acidities."* was changed to *"The experiments were conducted in presence of*

*ammonium bisulfate particles or ammonium sulfate particles mixed with sulfuric acid to achieve different seed acidities."*

Page 3, Line 92: "*$O_3$ was injected at the beginning of the experiments and ≈ 26 ppbv of TME was introduced into the aerosol chamber in steps of 15 minutes."* was changed to *"It was injected at the beginning of the experiments and ≈ 26 ppbv of TME was introduced into the aerosol chamber in steps of 15 minutes."*

Page 4, Line 96: *"The consumption of the precursor compound (ΔHC) was monitored over the reaction time of 90 minutes with a proton-transfer-reaction time-of-flight mass spectrometer (PTR-TOF MS)."* was changed to *"The consumption of the precursor compound (ΔHC) was monitored over a reaction time of 90 minutes with a proton-transfer-reaction time-of-flight mass spectrometer (PTR-TOF MS; 8000, IONICON Analytik, Innsbruck, Germany)."*

Page 4, Line 98: *"The volume size distribution of the seed particles was measured with a scanning mobility particle sizer (SMPS)."* was changed to *"The volume size distribution of the seed particles was measured with a scanning mobility particle sizer (SMPS; 3010, TSI, USA)."*

Page 4, Line 100: *"To collect the particle phase after the experiments 1.2 m³ of the chamber volume was sampled on a PTFE filter (borosilicate glass fiber filter coated with fluorocarbon, 47 mm in diameter, PALLFLEX T60A20, PALL, NY, USA) connected to a denuder (URG-2000-30B5, URG Corporation, Chapel Hill, NC, USA) (Kahnt et al., 2011) to avoid artefacts caused by adsorption of gas-phase organic compounds onto the filter."* was changed to *"To collect the particle phase after the experiments, 1.2 m³ of the chamber volume was sampled on a PTFE filter (borosilicate glass fiber filter coated with fluorocarbon, 47 mm in diameter, PALLFLEX T60A20, PALL, NY, USA) connected to a XAD-4 coated denuder (URG-2000-30B5, URG Corporation, Chapel Hill, NC, USA; Kahnt et al., 2011) to avoid artefacts caused by adsorption of gas-phase organic compounds onto the filter."*

Page 4, Line 109: The sentence *"Filter samples from the same experiments were used for the optimisation of the respective method parameters."* was added.

Page 4, Line 115: The paragraph *"Noticeably, the extraction efficiency was not investigated in the present study, thus it is not known. Water was used as extracting reagent, because organic solvents like methanol and acetonitrile have lower boiling points than water, thus lower heating temperatures can be applied for the decomposition of the oligomers. Besides this, organic solvents like methanol and acetonitrile are miscible with dichloromethane, thus an*

*extraction of the derivatised methylglyoxal with dichloromethane prior the injection into GC/MS would not be possible."* was included in the manuscript.

Page 4, Line 126: *"After the derivatisation was completed the extracts were allowed to cool down to room temperature."* was changed to *"After the derivatisation was complete, the extracts were allowed to cool down to room temperature."*

Page 4, Line 135: *"1 µL of the organic phase was used for GC/MS analysis and the measurement was repeated for three times to ensure reliable GC/MS signals."* was changed to *"1 µL of the organic phase was injected into GC/MS for analysis. The measurements were repeated for three times to ensure reliable GC/MS signals."*

Page 5, Line 146: The sentence *"The temperature of 230°C was held constant for 1 minute and ended with 320°C for 10 minutes."* was changed to *"The temperature of 230°C was held constant for 1 minute and ended with 320°C for 10 minutes, thus the method has a run time of 36 minutes."*

Page 5. Line 161: *"The influence of the heating time was examined with PTFE filters which were sampled after the OH-radical oxidation of 1,3,5-TMB at RH = 75% in the presence of $NH_4HSO_4$ seed particles (experiment #3)."* was changed to *"The influence of the heating time was examined with PTFE filters which were sampled after OH-radical oxidation of 1,3,5-TMB at RH = 75% in the presence of $NH_4HSO_4$ seed particles (experiment #3)."*

Page 5, Line 162: *"To investigate the effect of the heating time on the cleavage of the oligomeric compounds, the aqueous filter extracts (extract 1) were acidified to pH = 1 and heated to 100°C for 15 – 48 hours."* was changed to *"To investigate the effect of the heating time on the decomposition of the heat-decomposable oligomeric compounds, the aqueous filter extracts (extract 1) were acidified to pH = 1 and heated to 100°C for 15 – 48 hours."*

Page 5, Line 164: *"The results were compared to the unheated aqueous filter extract (extract 2) to determine the increase of methylglyoxal concentration due to decomposition of the oligomer."* was changed to *"The results were compared to the unheated aqueous filter extracts (extract 2) to determine the increase of methylglyoxal concentration due to decomposition of the heat-decomposable oligomers."*

Page 5, Line 169: *"After 24 hours, methylglyoxal concentration was about six times higher (c = 1.82 ± 0.14 µmol $L^{-1}$) in comparison to the unheated filter extract (c = 0.29 ± 0.01 µmol $L^{-1}$)."* was changed to *"The methylglyoxal concentration was about six*

*times higher (c = 1.82 ± 0.14 µmol L$^{-1}$) in comparison to the unheated filter extract (c = 0.29 ± 0.01 µmol L$^{-1}$).”*

Page 5, Line 173: *“A methylglyoxal concentration of c = 5.32 ± 0.05 µmol L$^{-1}$ was found which corresponds to a recovery of ≈ 85%.”* was changed to *“A methylglyoxal concentration of c = 5.32 ± 0.05 µmol L$^{-1}$ was found, which corresponds to a recovery of ≈ 85%.”*

Page 6, Line 182: *“Lower methylglyoxal concentration of sample b might be caused by an incomplete derivatisation due to the immediate measurement of the filter extract after 15 hours heating time.”* was changed to *“The lower methylglyoxal concentration of sample b might be caused by an incomplete derivatisation due to the immediate measurement of the filter extract after 15 hours heating time.”*

Page 6, Line 210: *“An increasing pH leads to a lower methylglyoxal concentration which can be observed for filter samples (Fig. 2a) as well as for methylglyoxal standard solution (Fig. 2b).”* was changed to *“An increasing pH leads to a lower methylglyoxal concentration, which can be observed for filter samples (Fig. 2a) as well as for the methylglyoxal standard solution (Fig. 2b).”*

Page 6, Line 213: *“As no influence of the pH on the PFBHA derivatisation reaction is reported by Rodigast et al., 2015, it can be concluded that the effect of the pH is connected to thermal decomposition of the oligomeric compounds.”* was changed to *“No influence of the pH on the PFBHA derivatisation reaction was reported by Rodigast et al. (2015) indicating the effect of the pH is connected to thermal decomposition of the heat-decomposable oligomeric compounds.”*

Page 7, Line 218: *“To examine the effect of the heating temperature, filter samples of experiment #2 were used.”* was changed to *“The effect of the heating temperature was examined with filter samples of experiment #2.”*

Page 7, Line 223: *“In comparison to the filter, which was neither acidified nor heated, the concentration was increased by a factor of two if the extract was heated to 100°C.”* was changed to *“In comparison to the filter, which was neither acidified nor heated, the concentration increased by a factor of two if the extract was heated to 100°C.”*

Page 7, Line 230: *“Noticeably, among the investigated filter samples no other carbonyl compounds showed an increase after thermal decomposition indicating that oligomers present in the particle phase of 1,3,5-TMB oxidation are solely methylglyoxal oligomers and/or oligomers of other carbonyl compounds need different conditions for decomposition.”* was

changed to *"According to the literature studies other carbonyl compounds can be expected as particle-phase products, e.g., propionaldehyde (Cocker et al., 2001), glyoxal (Cocker et al., 2001; Huang et al., 2015), 2-methyl-4-oxo-2-pentenal (Healy et al., 2008; Huang et al., 2014), glycolaldehyde (Cocker et al., 2001) and 3,5-dimethylbenzaldehyde (Huang et al., 2014). Noticeably, no carbonyl compounds other than methylglyoxal were identified, which showed an increase after thermal decomposition."*

Page 7, Line 239: *"Healy et al., 2008 determined SOA yields ($Y_{SOA}$) of 1,3,5-TMB photooxidation in a range of 4.5 – 8.3%."* was changed to *"Healy et al. (2008) determined SOA yields ($Y_{SOA}$) of 1,3,5-TMB photooxidation ranging from 4.5 to 8.3%."*

Page 7, Line 240: *"Further studies determined $Y_{SOA}$ between 0.29% and 15.6% (Table 4) under variation of the [HC]/[$NO_x$] ratio and concluded that SOA formation is enhanced at low $NO_x$ mixing ratios (Sato et al., 2012; Kleindienst et al., 1999; Baltensperger et al., 2005; Wyche et al., 2009; Odum et al., 1997; Paulsen et al., 2005; Cocker et al., 2001)."* was changed to *"Further studies determined $Y_{SOA}$ between 0.29% and 15.6% (Table 4) under variation of the [HC]/[$NO_x$] ratio concluding SOA formation is enhanced at low $NO_x$ mixing ratios (Sato et al., 2012; Kleindienst et al., 1999; Baltensperger et al., 2005; Wyche et al., 2009; Odum et al., 1997; Paulsen et al., 2005; Cocker et al., 2001)."*

Page 7, Line 244: *"The SOA yields ($Y_{SOA}$) were also determined in the present study for all conducted experiments based on the ratio of $\Delta M$ to $\Delta HC$ (Table 2)."* was changed to *"The SOA yields were also determined in the present study for all conducted experiments based on the ratio of $\Delta M$ to $\Delta HC$ (Table 2)."*

Page 7, Line 246: The sentence *"Noticeably, the SOA yields of 1,3,5-TMB are not corrected for the wall loss to the surface of the aerosol chamber, which might have an influence on the reported SOA yields."* was included.

Page 7, Line 251: *"These products condensate on the pre-existing seed particles resulting in the immediate particle growth observed in Fig. 4a."* was changed to *"These products condense on the pre-existing seed particles resulting in the immediate particle growth observed in Fig. 4a."*

Page 8, Line 259: The paragraph *"The RH value can have an influence on the phase state of the particles while the phase state has an effect on the partitioning of the compounds into the particles and the particle-phase reactions (Ziemann, 2010; Saukko et al., 2012). In the present study the particles are a mixture of inorganic and partitioned organic compounds, thus the*

*phase state is not known but it might be possible that the phase state influences ΔM and the SOA yields.”* was added to the manuscript.

Page 8, Line 269: *“Higher RH resulted in lower $Y_{SOA}$ between 4 and 5%.”* was changed to *“Higher RH values resulted in lower $Y_{SOA}$ between 4 and 5%.”*

Page 8, Line 276: *“This is also supported by Saxena and Hildemann, 1996 which found an enhanced partitioning of organic compounds with several hydroxyl groups at higher LWCs of the particles.”* was changed to *“This is also supported by Saxena and Hildemann (1996), which found an enhanced partitioning of organic compounds with several hydroxyl groups at higher LWCs of the particles.”*

Page 8, Line 279: *“Additionally, the formation of oligomeric compounds can be enhanced at lower RH values resulting in higher $Y_{SOA}$ due to the increasing conversion of the monomeric building blocks and their enhanced partitioning into the particle phase.”* was changed to *“Additionally, it can be speculated that the formation of oligomeric compounds can be enhanced at lower RH values resulting in higher $Y_{SOA}$ due to the increasing conversion of the monomeric building blocks and their enhanced partitioning into the particle phase.”*

Page 9, Line 305: *“The method is based on the thermal decomposition of the methylglyoxal oligomers.”* was changed to *“ The method is based on the thermal decomposition of the heat-decomposable methylglyoxal oligomers into monomers.”*

Page 9, Line 308: *“An oligomer fraction of ≈ 2 up to ≈ 8% was observed.”* was changed to *“An average oligomer fraction of ≈ 2% up to ≈ 8% was observed.”*

Page 9, Line 314: *“In Table 5 the pH of the seed particles were calculated with E-AIM.”* was changed to *“In Table 5 the pH of the seed particles was calculated with E-AIM.”*

Page 9, Line 321: *“Oligomerisation via acetal/hemiacetal formation occurs under water loss (Yasmeen et al., 2010).”* was changed to *“Oligomerisation via acetal/hemiacetal formation occurs under a reversible water loss (Yasmeen et al., 2010).”*

Page 9, Line 322: *“Higher RH in the aerosol chamber LEAK leads to higher LWCs in the seed particles (Table 5). With higher LWC in the particles the chemical equilibrium shifts towards the precursor compound resulting in a lower methylglyoxal oligomer fraction (Kalberer et al., 2004; Liggio et al., 2005).”* was changed to *“As higher RH values in the aerosol chamber LEAK leads to higher LWCs of the seed particles (Table 5) the chemical equilibrium of the*

*reaction shifts towards the precursor compound resulting in a lower methylglyoxal oligomer fraction (Kalberer et al., 2004; Liggio et al., 2005)."*

Page 9, Line 325: *"The pH of NH4HSO4 particles decreases with decreasing RH (Table 5). Acetal/hemiacetal formation is an acid-catalysed reaction and thus oligomer formation might be enhanced under dry conditions due to a lower pH (Liggio et al., 2005)."* was changed to *"In addition, the pH of NH4HSO4 particles decreases with decreasing RH (Table 5) thus acid-catalysed acetal/hemiacetal formation might be enhanced under dry conditions due to a lower pH (Liggio et al., 2005)."*

Page 9, Line 329: *"As it was mentioned, aldol condensation can be assumed as the favored accretion reaction (Yasmeen et al., 2010)."* was changed to *"As it was mentioned, aldol condensation can be assumed as the favored accretion reaction under these conditions (Yasmeen et al., 2010)."*

Page 10, Line 348: *"Nevertheless, it can be expected, that with higher LWC of the seed particles and thus with a higher reaction volume, the absolute amount of methylglyoxal in the particle phase might increases, but not its particle-phase concentrations."* was changed to *"Nevertheless, it can be expected, that with higher LWCs of the seed particles and thus with a higher reaction volume, the absolute amount of methylglyoxal in the particle phase might increase, but not its particle-phase concentration."*

Page 10, Line 358: *"In summary, the present study provides a reliable quantification method of methylglyoxal oligomers formed by 1,3,5-TMB oxidation."* was changed to *"In summary, the present study provides a reliable quantification method for heat-decomposable methylglyoxal oligomers formed by 1,3,5-TMB oxidation."*

Page 13, Line 502: The reference Huang et al., 2014 *"Huang, M. Q., Hu, C. J., Guo, X. Y., Gu, X. J., Zhao, W. X., Wang, Z. Y., Fang, L., and Zhang, W. J.: Chemical composition of gas and particle-phase products of OH-initiated oxidation of 1,3,5-trimethylbenzene, Atmospheric Pollution Research, 5, 73-78, 10.5094/APR.2014.009, 2014."* was included in the manuscript

Page 14, Line 549: The reference *"Lim, Y. B., and Turpin, B. J.: Laboratory evidence of organic peroxide and peroxyhemiacetal formation in the aqueous phase and implications for aqueous OH, Atmospheric Chemistry and Physics, 15, 12867-12877, 10.5194/acp-15-12867-2015, 2015."* was included.

Page 14, Line 559: The reference was changed to *"Müller, L., Reinnig, M. C., Warnke, J., and Hoffmann, T.: Unambiguous identification of esters as oligomers in secondary organic aerosol*

*formed from cyclohexene and cyclohexene/alpha-pinene ozonolysis, Atmospheric Chemistry and Physics, 8, 1423-1433, 2008.”*

Page 14, Line 562: The reference was changed to *“Müller, M., Graus, M., Wisthaler, A., Hansel, A., Metzger, A., Dommen, J., and Baltensperger, U.: Analysis of high mass resolution PTR-TOF mass spectra from 1,3,5-trimethylbenzene (TMB) environmental chamber experiments, Atmospheric Chemistry and Physics, 12, 829-843, 10.5194/acp-12-829-2012, 2012.”*

Page 14, Line 590: The reference Paulot et al., 2009 *“Paulot, F., Crounse, J. D., Kjaergaard, H. G., Kurten, A., St Clair, J. M., Seinfeld, J. H., and Wennberg, P. O.: Unexpected Epoxide Formation in the Gas-Phase Photooxidation of Isoprene, Science, 325, 730-733, 10.1126/science.1172910, 2009.”* was included in the manuscript.

Page 15, Line 616: The reference Sauko et al., 2012 *“Saukko, E., Lambe, A. T., Massoli, P., Koop, T., Wright, J. P., Croasdale, D. R., Pedernera, D. A., Onasch, T. B., Laaksonen, A., Davidovits, P., Worsnop, D. R., and Virtanen, A.: Humidity-dependent phase state of SOA particles from biogenic and anthropogenic precursors, Atmospheric Chemistry and Physics, 12, 7517-7529, 10.5194/acp-12-7517-2012, 2012.”* was included in the manuscript.

Page 15, Line 654: The reference Surratt et al., 2010 *“Surratt, J. D., Chan, A. W. H., Eddingsaas, N. C., Chan, M. N., Loza, C. L., Kwan, A. J., Hersey, S. P., Flagan, R. C., Wennberg, P. O., and Seinfeld, J. H.: Reactive intermediates revealed in secondary organic aerosol formation from isoprene, Proceedings of the National Academy of Sciences of the United States of America, 107, 6640-6645, 10.1073/pnas.0911114107, 2010.”* was included in the manuscript.

Page 16, Line 692: The reference Ziemann, 2010 *“Ziemann, P. J.: Atmospheric chemistry Phase matters for aerosols, Nature, 467, 797-798, 2010.”* was included in the manuscript.

[revised manuscript text omitted]

---

## Author Response (AR2)

**Responses to Reviewer Comments:**

We thank the reviewer for the constructive comments. The manuscript is revised based on the suggestions made and detailed responses to the reviewer are addressed as follows.

Reviewer #1

The revised version addresses the concerns of the reviewers, with two notable exceptions. First, the authors still do not correct for wall losses, but now clearly state this and mention that this will affect the reported yields. I'm willing to accept that, as long as they state the direction of this effect (see comment regarding line 272 below).

Author´s comment

The wall loss might have an influence due to partitioning of the precursor compounds and products to the wall of the aerosol chamber resulting in lower SOA yields. To mention this in the manuscript the sentence (Line 270) was changed to "*Notably, the SOA yields of 1,3,5-TMB are not corrected for the wall loss to the surface of the aerosol chamber, which might possibly lead to an underestimation of the reported SOA yields.*".

Second, the authors make an argument that particle phase may influence the results, but cannot be predicted. This is generally true, but in my opinion some of the information provided in their responses to reviewers' comments would be very helpful to readers if included in the manuscript. Specifically, the differences between the inorganic components ammonium sulfate (solid at all RH used in the study) and ammonium bisulfate (liquid at all RH used in the study except <2% RH) are significant and should be mentioned. The authors could also mention the caveats (also given in their response to reviewers' comments) that the organic phase could lower the DRH of ammonium sulfate and cause it to deliquesce at 75% RH (but not likely at 0 or 50% RH), and that the formation of significant quantities of methylglyoxal oligomers could cause liquid aerosol particles (or liquid organic particle phases) to become semi-solid. Based on the work of Bertram et al.,(1) I think the authors can confidently state that at 0 and 50% RH, ammonium sulfate will be present in crystalline form along with an organic phase (of unknown viscosity), while at 50 and 75% RH crystalline ammonium bisulfite will not be present. If the authors included seed particle loadings, readers could interpret the ammonium sulfate data according to the organic / sulfate ratio framework of Bertram et al.(1)

Author´s comment

To clarify this some more information has been included in the manuscript.

Line 281: *"The RH value can have an influence on the phase state of the particles which, in turn, has an effect on the partitioning of the compounds into the particles as well as on particle-phase reactions (Ziemann, 2010; Saukko et al., 2012). The deliquescence RH of $(NH_4)_2SO_4$ and $NH_4HSO_4$ seed particles are known from literature to be 79% and 39%, respectively (Cziczo et al., 1997). Thus, pure $NH_4HSO_4$ seed particles are liquid at the applied RH values of 50% and 75% and solid at RH = 0% while $(NH_4)_2SO_4$ seed particles are solid over the whole RH range. Nevertheless, in the present study the particles are a mixture of inorganic and partitioned organic compounds. Organic compounds might change the phase state of the particles (Bertram et al., 2011) due to an influence on the deliquescence point (Andrews and Larson, 1993; Lightstone et al., 2000) as well as the hygroscopic behaviour of the particles (Lightstone et al., 2000; Prenni et al., 2003; Chen and Lee, 1999). Virtanen et al., 2010 postulated that particles are in an amorphous solid state if oligomeric compounds are present in the particles. Thus, it can be speculated that the particles in the present study containing a fraction of up to 8% of oligomeric compounds might be in an amorphous solid phase state. It was assumed, that the further reactions in the particle phase might be inhibited in solid particles (Saukko et al., 2012) thus, further oligomerisation or other reactions might become less effective after a certain fraction of oligomers exist in the particles. In addition, the partitioning of methylglyoxal monomers can be inhibited into solid particles (Saukko et al., 2012), which then might also lead to lower oligomer fractions in SOA. Thus, it might be possible that the phase state influences $\Delta M$ and the SOA yields."*

Accordingly, the following references were included in the manuscript:

*"Andrews, E., and Larson, S. M.: Effect of surfactant layers on the size changes of aerosol-particles as a function of relative-humidity, Environmental Science & Technology, 27, 857-865, 10.1021/es00042a007, 1993."*

*"Bertram, A. K., Martin, S. T., Hanna, S. J., Smith, M. L., Bodsworth, A., Chen, Q., Kuwata, M., Liu, A.; You, Y., and Zorn, S. R.: Predicting the relative humidities of liquid-liquid phase separation, efflorescence, and deliquescence of mixed particles of ammonium sulfate, organic material, and water using the organic-to-sulfate mass ratio of the particle and the oxygen-to-carbon elemental ratio of the organic component, Atmospheric Chemistry and Physics, 11, 10995-11006, 10.5194/acp-11-10995-2011, 2011."*

*"Chen, Y. Y., and Lee, W. M. G.: Hygroscopic properties of inorganic-salt aerosol with surface-active organic compounds, Chemosphere, 38, 2431-2448, 10.1016/s0045-6535(98)00436-6, 1999."*

*"Cziczo, D. J., Nowak, J. B., Hu, J. H., and Abbatt, J. P. D.: Infrared spectroscopy of model tropospheric aerosols as a function of relative humidity: Observation of deliquescence and crystallization, Journal of Geophysical Research-Atmospheres, 102, 18843-18850, 10.1029/97jd01361, 1997."*

*"Lightstone, J. M., Onasch, T. B., Imre, D., and Oatis, S.: Deliquescence, efflorescence, and water activity in ammonium nitrate and mixed ammonium nitrate/succinic acid microparticles, Journal of Physical Chemistry A, 104, 9337-9346, 10.1021/jp002137h, 2000."*

*"Prenni, A. J., De Mott, P. J., and Kreidenweis, S. M.: Water uptake of internally mixed particles containing ammonium sulfate and dicarboxylic acids, Atmospheric Environment, 37, 4243-4251, 10.1016/s1352-2310(03)00559-4, 2003."*

*"Virtanen, A., Joutsensaari, J., Koop, T., Kannosto, J., Yli-Pirila, P., Leskinen, J., Makela, J. M., Holopainen, J. K., Poschl, U., Kulmala, M., Worsnop, D. R., and Laaksonen, A.: An amorphous solid state of biogenic secondary organic aerosol particles, Nature, 467, 824-827, 10.1038/nature09455, 2010."*

Other comments:

Line 75: The statement "The method is applicable for all oligomeric compounds, which can be decomposed into methylglyoxal monomers" is misleading as currently stated. It would be clearer to state "The method is applicable for any oligomeric compound which can be decomposed into methylglyoxal monomers."

Author´s comment

The author´s agree with the reviewer and changed the sentence to *"The method is applicable for any oligomeric compounds, which can be decomposed into methylglyoxal monomers during the heating process at a temperature of 100 °C."*

Line 133: Technically, the filter extract that was not heated is not measuring only methylglyoxal monomers, but also any compound that can be hydrolyzed at room temperature into methylglyoxal monomers. This likely includes some acetal oligomers. Also, the authors should make clear at some point in the manuscript that "methylglyoxal monomers" includes singly and doubly hydrated methylglyoxal.

Author´s comment

The author´s agree with the reviewer. The sentence was changed to *"The second half of the filters was used to quantify monomeric methylglyoxal including also singly and doubly hydrated methylglyoxal, which can be hydrolysed at room temperature into monomers."*

Line 394: There also might be oligomers originating from methylglyoxal monomers that are not heat-decomposible back to methylglyoxal monomers, which would also not be determined in the present study.

To clarify this, the sentence was changed to *"[…] thus there might be not heat-decomposable methylglyoxal oligomers or oligomers originating from other monomers […]"*

Technical corrections:

Line 20: The phrase "emitted hydrocarbons contributing with up to …" would be clearer if written as "emitted hydrocarbons, contributing up to …"

The sentence was changed to *"[…] fraction of the emitted hydrocarbons, contributing up to 52% […]"*

Line 24: "low-volatile" should be "low-volatility"

The sentence was changed to *"[…] leads to low-volatility oxidation products […]"*

Line 127 and 271: I think that "Noticeably" should be "Notably"

*"Noticeably"* was changed to *"Notably"* in Line 127 and 271.

Line 148: it would be helpful to give essential parameters of this method here.

*"The filters were prepared according to the method described by Rodigast et al. (2015)."* was changed to *"The filters were prepared according to the method described by Rodigast et al. (2015), thus 300 µL of an aqueous PFBHA solution (5 mg mL$^{-1}$) was added to the filter extracts. After a derivatisation time of 24 hours at room temperature the derivatised methylglyoxal monomers were extracted."*

Line 272: The phrase " …might have an influence on the reported SOA yields." would be more informative as "…might exert a downward influence on the reported SOA yields."

The sentence (Line 270) was changed to *"Notably, the SOA yields of 1,3,5-TMB are not corrected for the wall loss to the surface of the aerosol chamber, which might possibly lead to an underestimation of the reported SOA yields."*.

Line 299: The phrase "… controversially discussed in the literature" could be changed to a single word "… controversial"

The sentence was changed to *"[…] on SOA formation is controversial […]"*.

Table 4: It would be good to include the results of the present study in Table 4.

The results of the present study were included in Table 4.

**The following changes were made to the manuscript**

Line 20: *"Aromatic compounds represent a large fraction of the emitted hydrocarbons contributing with up to 52% to the total non-methane hydrocarbon mass at an industrial dominated site in China (Liu et al., 2008)."* was changed to *"Aromatic compounds represent a large fraction of the emitted hydrocarbons, contributing up to 52% to the total non-methane hydrocarbon mass at an industrial dominated site in China (Liu et al., 2008)."*

Line 23: *"The gas-phase oxidation of 1,3,5-TMB leads to low-volatile oxidation products, which partition into the particle phase and form secondary organic aerosol (SOA)."* was changed to *"The gas-phase oxidation of 1,3,5-TMB leads to low-volatility oxidation products, which partition into the particle phase and form secondary organic aerosol (SOA)."*

Line 75: *"The method is applicable for all oligomeric compounds, which can be decomposed into methylglyoxal monomers during the heating process at a temperature of 100 °C."* was changed to *"The method is applicable for any oligomeric compounds, which can be decomposed into methylglyoxal monomers during the heating process at a temperature of 100 °C."*

Line 127: *"Noticeably, the extraction efficiency was not investigated in the present study, thus it is not known."* was changed to *"Notably, the extraction efficiency was not investigated in the present study, thus it is not known."*

Line 146: *"The second half of the filters was used to quantify monomeric methylglyoxal."* was changed to *"The second half of the filters was used to quantify monomeric methylglyoxal*

*including also singly and doubly hydrated methylglyoxal, which can be hydrolysed at room temperature into monomers."*

Line 146: *"The filters were prepared according to the method described by Rodigast et al. (2015)."* was changed to *"The filters were prepared according to the method described by Rodigast et al. (2015), thus 300 µL of an aqueous PFBHA solution (5 mg mL$^{-1}$) was added to the filter extracts. After a derivatisation time of 24 hours at room temperature the derivatised methylglyoxal monomers were extracted."*

Line 270; *"Noticeably, the SOA yields of 1,3,5-TMB are not corrected for the wall loss to the surface of the aerosol chamber, which might have an influence on the reported SOA yields."* was changed to *"Notably, the SOA yields of 1,3,5-TMB are not corrected for the wall loss to the surface of the aerosol chamber, which might possibly lead to an underestimation of the reported SOA yields."*

*Line 281: " The RH value can have an influence on the phase state of the particles while the phase state has an effect on the partitioning of the compounds into the particles and the particle-phase reactions (Ziemann, 2010; Saukko et al., 2012). In the present study the particles are a mixture of inorganic and partitioned organic compounds, thus the phase state is not known but it might be possible that the phase state influence ΔM and the SOA yields."* was changed to *"The RH value can have an influence on the phase state of the particles which, in turn, has an effect on the partitioning of the compounds into the particles as well as on particle-phase reactions (Ziemann, 2010; Saukko et al., 2012). The deliquescence RH of $(NH_4)_2SO_4$ and $NH_4HSO_4$ seed particles are known from literature to be 79% and 39%, respectively (Cziczo et al., 1997). Thus, pure $NH_4HSO_4$ seed particles are liquid at the applied RH values of 50% and 75% and solid at RH = 0% while $(NH_4)_2SO_4$ seed particles are solid over the whole RH range. Nevertheless, in the present study the particles are a mixture of inorganic and partitioned organic compounds. Organic compounds might change the phase state of the particles (Bertram et al., 2011) due to an influence on the deliquescence point (Andrews and Larson, 1993; Lightstone et al., 2000) as well as the hygroscopic behaviour of the particles (Lightstone et al., 2000; Prenni et al., 2003; Chen and Lee, 1999). Virtanen et al., 2010 postulated that particles are in an amorphous solid state if oligomeric compounds are present in the particles. Thus, it can be speculated that the particles in the present study containing a fraction of up to 8% of oligomeric compounds might be in an amorphous solid phase state. It was assumed, that the further reactions in the particle phase might be inhibited*

*in solid particles (Saukko et al., 2012) thus, further oligomerisation or other reactions might become less effective after a certain fraction of oligomers exist in the particles. In addition, the partitioning of methylglyoxal monomers can be inhibited into solid particles (Saukko et al., 2012), which then might also lead to lower oligomer fractions in SOA. Thus, it might be possible that the phase state influences ΔM and the SOA yields."*

Line 298: *" The influence of RH on SOA formation is controversially discussed in the literature (Hennigan et al., 2008; Fick et al., 2003; Edney et al., 2000; Saxena and Hildemann, 1996; Baker et al., 2001; Hasson et al., 2001, Cocker et al., 2001)"* was changed to *" The influence of RH on SOA formation is controversial (Hennigan et al., 2008; Fick et al., 2003; Edney et al., 2000; Saxena and Hildemann, 1996; Baker et al., 2001; Hasson et al., 2001, Cocker et al., 2001)."*

[revised manuscript text omitted]